# EXPLORING ADVERSARIAL ROBUSTNESS OF GRAPH NEURAL NETWORKS IN DIRECTED GRAPHS

## ABSTRACT

Existing research on robust Graph Neural Networks (GNNs) focuses predominantly on undirected graphs, neglecting the trustworthiness inherent in directed graphs. This work analyzes the limitations of existing approaches from both attack and defense perspectives, and we present an exploration of the adversarial robustness of GNNs in directed graphs. Specifically, we first introduce a new and more realistic directed graph attack setting to overcome the limitations of existing attacks. Then we propose a simple and effective message-passing framework as a plug-in layer to enhance the robustness of GNNs while avoiding a false sense of security. Our findings demonstrate that the profound trust implications offered by directed graphs can be harnessed to bolster the robustness and resilience of GNNs significantly. When coupled with existing defense strategies, this framework achieves outstanding clean accuracy and state-of-the-art robust performance against both transfer and adaptive attacks.

## 1 INTRODUCTION

Graph neural networks (GNNs) have emerged to be a promising approach for learning feature representations from graph data, owing to their ability to capture node features and graph topology information through message-passing frameworks (Ma & Tang, 2020; Hamilton, 2020). However, extensive research has revealed that GNNs are vulnerable to adversarial attacks (Dai et al., 2018; Jin et al., 2021; Wu et al., 2019; Zügner et al., 2018b; Zügner & Günnemann, 2019). Even slight perturbations in the graph structure can lead to significant performance deterioration. Despite the existence of numerous defense strategies, their effectiveness has been questioned due to a potential false sense of robustness against transfer attacks (Mujkanovic et al., 2022). In particular, a recent study (Mujkanovic et al., 2022) demonstrated that existing robust GNNs are much less robust when facing stronger adaptive attacks. In many cases, these models even underperform simple multi-layer perceptions (MLPs) that disregard graph topology information, indicating the failure of GNNs in the presence of adversarial attacks. As existing research fails to deliver satisfying robustness, new strategies are needed to effectively enhance the robustness of GNNs.

As evident from the literature (Jin et al., 2021; Dai et al., 2022; Mujkanovic et al., 2022), most existing research on the attack and defense of GNNs focuses on undirected graphs. From the *attack perspective*, existing attack algorithms (Dai et al., 2018; Jin et al., 2021; Wu et al., 2019; Zügner et al., 2018b; Zügner & Günnemann, 2019) flip both directions of an edge (out-link and in-link) when it is selected, which could be unrealistic in many real-world scenarios. For instance, in a social network as shown in Figure 1, it is relatively easy to create many fake users and orchestrate large-scale link spam (i.e., in-links) targeting specific users (Alkhalil et al., 2021). However, hacking into the accounts of those target users and manipulating their following behaviors (i.e., out-links) are considerably more difficult (Gohel, 2015). From the *defense perspective*, most robust GNNs (Mujkanovic et al., 2022) convert the directed graphs to undirected ones through symmetrization, leading to the loss of valuable directional information. Despite the existence of directed GNNs, their adversarial robustness is largely unexplored.

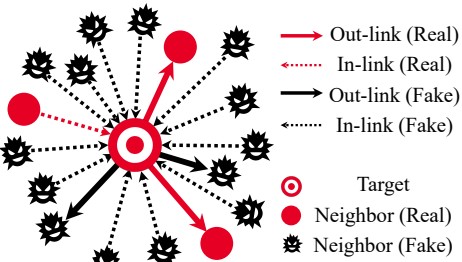

| | |
|---|---|
| →  | Out-link (Real) |
| ←···· | In-link (Real) |
| → | Out-link (Fake) |
| ←···· | In-link (Fake) |
| ⊙ | Target |
| ● | Neighbor (Real) |
| ⊛ | Neighbor (Fake) |

Figure 1: Large-scale link spam attack.

On the contrary to undirected graphs, many graphs in real-world applications such as citation networks (Radicchi et al., 2011), social networks (Robins et al., 2009), and web networks (Kleinberg et al., 1999) are naturally directed. The link directions in directed graphs inherently imply trustworthiness: *out-links are usually more trustworthy than in-links for a target node* (Page et al., 1998; Kamvar et al., 2003; Gyongyi et al., 2004). This is because out-links are usually formed by active behaviors such as citing a paper in citation networks, following a user on social media, pointing to a page on Web, or making payment to an account in transaction networks. Therefore, it is practically more challenging to attack out-links than in-links of target nodes. Regardless of the trustworthiness offered by directed graphs, the fact that most existing attacks and defenses are limited to undirected graphs leaves the robustness and trustworthiness of GNNs in directed graphs underexplored.

To address the aforementioned research gap, we propose to explore adversarial robustness in directed graphs from both attack and defense perspectives. From the *attack perspective*, we introduce a weaker but more realistic attack setting (Section 2) that differentiates out-link and in-link attacks while imposing certain restrictions on out-link attacks to reflect the practical challenges of manipulating out-links. From the *defense perspective*, we propose a simple yet effective message-passing layer to protect GNNs against adversarial attacks. Our contributions can be summarized as follows:

- We analyze the limitations of existing research on the attacks and defenses in undirected graphs, and introduce Restricted Directed Graph Attack (RDGA), a new and more realistic adversarial graph attack setting for directed graphs.

- We propose a simple yet effective Biased Bidirectional Random Walk (BBRW) message-passing layer that avoids the catastrophic failure we discover and substantially enhances the robustness of various GNN backbones as a plug-in layer.

- Our comprehensive comparison showcases that BBRW achieves outstanding clean accuracy and state-of-the-art robustness against both transfer and adaptive attacks. We provide detailed ablation studies to further understand the working mechanism of the proposed approach.

## 2 RESTRICTED DIRECTED GRAPH ATTACK

In this section, we first discuss the limitations of existing adversarial graph attack settings for undirected graphs and introduce a more realistic adversarial graph attack setting for directed graphs.

**Notations.** In this paper, we consider a directed graph $\mathcal{G} = (\mathcal{V}, \mathcal{E})$ with $|\mathcal{V}| = n$ nodes and $|\mathcal{E}| = m$ edges. The adjacency matrix of $\mathcal{G}$ is denoted as $\mathbf{A} \in \{0, 1\}^{n \times n}$. The feature matrix of $n$ nodes is denoted as $\mathbf{X} \in \mathbb{R}^{n \times d}$. The label matrix is denoted as $\mathbf{Y} \in \mathbb{R}^n$. The degree matrix of $\mathbf{A}$ is $\mathbf{D} = \text{diag}\,(d_1, d_2, ..., d_n)$, where $d_i = \sum_j \mathbf{A}_{ij}$ is the out-degree of node $i$. $f_\theta(\mathbf{A}, \mathbf{X})$ denotes the GNN encoder that extract features from $\mathbf{A}$ and $\mathbf{X}$ with network parameters $\theta$.

### 2.1 LIMITATIONS OF EXISTING ADVERSARIAL GRAPH ATTACK

Existing adversarial graph attacks mostly conduct undirected graph attacks that flip both directions (in-link and out-link) of an adversarial edge once being selected (Xu et al., 2019; Chen et al., 2018; Zügner et al., 2018b; Zügner & Günnemann, 2019). However, this common practice has some critical limitations. First, it is often impractical to attack both directions of an edge in graphs. For instance, flipping the out-links of users in social media platforms or financial systems usually requires hacking into their accounts to change their following or transaction behaviors, which can be easily detected by security countermeasures such as Intrusion Detection Systems (Bace et al., 2001). Second, the undirected graph attack setting does not distinguish the different roles of in-links and out-links, which fundamentally undermines the resilience of networks. For instance, a large-scale link spam attack targeting a user does not imply the targeted user fully trusts these in-links. But the link spam attack can destroy the feature of target nodes if being made undirected. Due to these limitations, existing graph attacks are not practical in many real-world applications, and existing defenses can not effectively leverage useful information from directed graphs.

### 2.2 RESTRICTED DIRECTED GRAPH ATTACK

To overcome the limitations of existing attack and defense research on GNNs, we propose Restricted Directed Graph Attack (RDGA), a more realistic graph attack setting that differentiates between

in-link and out-link attacks on target nodes while restricting the adversary's capability to execute out-link attacks on target nodes, which aligns with the practical challenges of manipulating out-links.

**Adversarial Capacity.** Mathematically, we denote the directed adversarial attack on the directed graph $\mathbf{A} \in \{0,1\}^{n \times n}$ as an asymmetric perturbation matrix $\mathbf{P} \in \{0,1\}^{n \times n}$. The adjacency matrix being attacked is given by $\tilde{\mathbf{A}} = \mathbf{A} + (\mathbf{1}\mathbf{1}^\top - 2\mathbf{A}) \odot \mathbf{P}$ where $\mathbf{1} = [1, 1, \ldots, 1]^\top \in \mathbb{R}^n$ and $\odot$ denotes element-wise product. $\mathbf{P}_{ij} = 1$ means flipping the edge $(i,j)$ (i.e., $\tilde{\mathbf{A}}_{ij} = 0$ if $\mathbf{A}_{ij} = 1$ or $\tilde{\mathbf{A}}_{ij} = 1$ if $\mathbf{A}_{ij} = 0$) while $\mathbf{P}_{ij} = 0$ means keeping the edge $(i,j)$ unchanged (i.e., $\tilde{\mathbf{A}}_{ij} = \mathbf{A}_{ij}$). The asymmetric nature of this perturbation matrix indicates the adversarial edges have directions so that one direction will not necessarily imply the attack from the opposite direction as in existing attacks.

Given the practical difficulty of attacking the out-links on the target nodes, we impose restrictions on the adversary's capacity for executing out-link attacks on target nodes. The Restricted Directed Graph Attack (RDGA) is given by $\tilde{\mathbf{A}} = \mathbf{A} + (\mathbf{1}\mathbf{1}^\top - 2\mathbf{A}) \odot (\mathbf{P} \odot \mathbf{M})$, where $\tilde{\mathbf{P}} = \mathbf{P} \odot \mathbf{M}$ denotes the restricted perturbation. When restricting the out-link of nodes $\mathcal{T}$ (e.g., the target nodes), the mask matrix is defined as $\mathbf{M}_{ij} = 0 \; \forall i \in \mathcal{T}, j \in \mathcal{N}$ and $\mathbf{M}_{ij} = 1$ otherwise.

**Attacking Algorithm.** The attacking process closely follows existing undirected graph attacks such as PGD attack (Xu et al., 2019), FGA (Mujkanovic et al., 2022), or Nettack (Zügner et al., 2018a), but it additionally considers different attacking budgets for in-links and out-links when selecting the edges as demonstrated in the adversarial capacity. Among the commonly used attacks, FGA (Mujkanovic et al., 2022), Nettack (Zügner et al., 2018a) and Metattack (Sun et al., 2020) employ greedy approaches and tend to provide relatively weaker attacks (Mujkanovic et al., 2022). Alternatively, PGD (Xu et al., 2019) derives a probabilistic perturbation matrix through gradient-based optimization and then samples the strongest perturbation from it. Since PGD attack exhibits the strongest attack as verified by our experiments in Appendix A.2, we majorly adopt PGD attack in our experiment and present the attack algorithm in Appendix A.1. In Section 4.3, we also study a more general RDGA that allows some portion of the attack budgets on targets' out-links where the masking matrix is partially masked.

## 3 Methodology: Robust GNNs in Directed Graphs

While the directed attack proposed in Section 2 is weaker (but more realistic) than existing undirected attacks due to the additional constraints, undirected GNNs will perform the same under both attacks since they lose directional information after symmetrization. In spite of this, it still offers unprecedented opportunities to design robust GNNs that distinguish the roles of in-links and out-links in directed graphs. In this section, we first discover the catastrophic failures of GNNs with directed random walk message passing. This motivates the design of simple and effective GNNs with biased bidirectional random walk message passing. We also provide a theoretical case study to understand the discovered catastrophic failures and the working mechanism of the proposed algorithm.

### 3.1 Catastrophic Failures of Directed Random Walk Message Passing

Due to the adversary's capacity constraint on out-link attacks, out-links are more reliable than in-links, which aligns better with real-world examples as demonstrated in Section 1 and Section 2. This motivates to first study directed random walk message passing (RW) that only aggregates node features from out-links: $\mathbf{X}^{l+1} = \mathbf{D}^{-1}\mathbf{A}\mathbf{X}^l$. We use two popular GNNs including GCN (Kipf & Welling, 2016) and APPNP (Gasteiger et al., 2018) as the backbone models and substitute their symmetric aggregation matrix $\mathbf{D}^{-\frac{1}{2}}\mathbf{A}_{\text{sym}}\mathbf{D}^{-\frac{1}{2}}$ as $\mathbf{D}^{-1}\mathbf{A}$, denoted as GCN-RW and APPNP-RW.

We evaluate the clean and robust node classification accuracy of these variants on the Cora-ML dataset under RDGA, following the experimental setting detailed in Section 4. It is worth emphasizing that while we transfer attacks from the surrogate model GCN as usual, we additionally test the robust performance of adaptive attacks which directly attack the victim model to avoid a potential false sense of robustness. The results in Table 1 provide the following insightful observations:

- In terms of clean accuracy, we have GCN > GCN-RW > MLP and APPNP > APPNP-RW > MLP. This indicates that both out-links and in-links in the clean directed graph provide useful graph topology information. Undirected GNNs (GCN and APPNP) achieve the best clean performance since both in-links and out-links are utilized through symmetrization.

Table 1: Classification accuracy (%) under transfer and adaptive attacks (Cora-ML)

| Method \ Budget | 0% | 25% | | 50% | | 100% | |
|---|---|---|---|---|---|---|---|
| | Clean | Transfer | Adaptive | Transfer | Adaptive | Transfer | Adaptive |
| MLP | 73.5±7.4 | 73.5±7.4 | 73.5±7.4 | 73.5±7.4 | 73.5±7.4 | 73.5±7.4 | 73.5±7.4 |
| GCN | 89.5±6.1 | 66.0±9.7 | 66.0±9.7 | 40.5±8.5 | 40.5±8.5 | 12.0±6.4 | 12.0±6.4 |
| GCN-RW | 86.5±6.3 | 86.5±6.3 | **52.0±8.1** | 86.5±6.3 | **28.0±4.6** | 86.5±6.3 | **10.5±5.7** |
| APPNP | 90.5±4.7 | 81.5±9.5 | 80.5±10.4 | 66.5±8.7 | 68.0±12.1 | 44.0±9.2 | 46.0±7.3 |
| APPNP-RW | 85.5±6.5 | 85.5±6.5 | **30.0±7.7** | 85.5±6.5 | **15.0±3.9** | 85.0±6.3 | **11.5±3.2** |

- Under transfer attacks, we have GCN-RW > GCN > MLP and APPNP-RW > APPNP > MLP. Transfer attacks barely impact GCN-RW and APPNP-RW since no out-link attack is allowed under RDGA setting and RW is free from the impact of in-link attacks. However, in-link attacks hurt GCN and APPNP badly due to the symmetrization operation.

- Although RW performs extremely well under transfer attacks, we surprisingly find that GCN-RW and APPNP-RW suffer from *catastrophic failures* under stronger adaptive attacks, and they significantly underperform simple MLP, which uncovers a severe *false sense of robustness*.

**Catastrophic Failures due to Indirect Attacks.** The catastrophic failures of GCN-RW and APPNP-RW under adaptive attacks indicate their false sense of robustness.

In order to understand this phenomenon and gain deeper insights, we perform statistical analyses on the adversary behaviors when attacking different victim models such as GCN and GCN-RW using attack budget 50% (Figure 2). Note that similar observations can be made under other attack budgets as shown in Appendix A.4. In particular, we separate adversarial links into different groups according to whether they directly connect target nodes or targets' neighbors. The yellow portion repre-

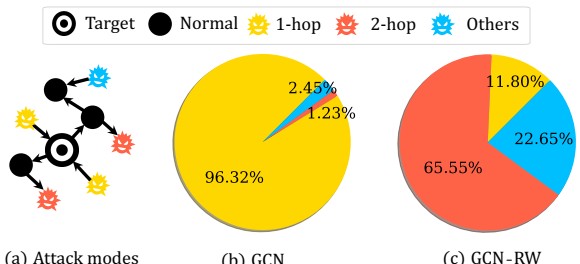

(a) Attack modes     (b) GCN     (c) GCN-RW

Figure 2: Adversary behaviors.

sents attacks by 1-hop neighbors on the target (direct in-link attacks); the red portion represents attacks by 2-hop neighbors on the target (indirect out-link attacks); and the blue portion represents other attacks. The distributions of adversarial links shown in Figure 2 indicate:

- When attacking GCN (Figure 2 (b)), the adversary majorly attacks the 1-hop in-links of target nodes using 96.32% perturbation budget, which badly hurts its performance since GCN replies on both in-links and out-links. However, the attack transferred from these two victim models barely impact GCN-RW that only trusts out-links.

- When attacking GCN-RW (Figure 2 (c)), the adversary can not manipulate the 1-hop out-links of target nodes under the restricted setting (RDGA). It does not focus on attacking the 1-hop in-links of target nodes either since these 1-hop in-links can not influence GCN-RW. Instead, the adversary tactfully identifies the targets' neighbors and conducts 2-hop out-link attacks through these neighbors using 65.55% budget. In other words, it focuses on attacking the out-linking neighbors of target nodes such that these neighbors can destroy the predictions of target nodes.

## 3.2   BIASED BIDIRECTIONAL RANDOM WALK MESSAGE PASSING

The study on the directed random walk message passing in Section 3.1 indicates that it is non-trivial to robustify GNNs using directed graphs, but it provides insightful motivations to develop a better approach. In this section, we propose a simple and effective approach with theoretical justification.

The systematic study on Section 3.1 offers two valuable lessons: **(1)** Both in-links and out-links provide useful graph topology information; **(2)** While out-links are more reliable than in-links, full trust in out-links can cause catastrophic failures and a false sense of robustness under adaptive attacks

due to the existence of indirect attacks. These lessons motivate us to develop a message-passing framework that not only fully utilizes the out-links and in-links information but also differentiates their roles. Importantly, it also needs to avoid a false sense of robustness under adaptive attacks.

To this end, we propose a Biased Bidirectional Random Walk (BBRW) Message Passing framework represented by the propagation matrix that balances the trust on out-links and in-links:

$$\tilde{\mathbf{A}}_\beta = \mathbf{D}_\beta^{-1}\mathbf{A}_\beta \quad \text{where} \quad \mathbf{D}_\beta = \mathbf{A}_\beta \mathbf{1}, \quad \mathbf{A}_\beta = \beta\mathbf{A} + (1-\beta)\mathbf{A}^\top.$$

$\mathbf{A}_\beta$ is the weighted sum of $\mathbf{A}$ and $\mathbf{A}^\top$ that combines the out-links (directed random walk) and in-links (inversely directed random walk), i.e., $\{\mathbf{A}_\beta\}_{ij} = \beta\mathbf{A}_{ij} + (1-\beta)\mathbf{A}_{ji}$. $\mathbf{D}_\beta$ is the out-degree matrix of $\mathbf{A}_\beta$. $\tilde{\mathbf{A}}_\beta$ denotes the random walk normalized propagation matrix that aggregates node features from both out-linking and in-linking neighbors. The bias weight $\beta \in [0,1]$ controls the relative trustworthiness of out-links compared with in-links. When $\beta = 1$, it reduces to RW that fully trusts out-links. But RW suffers from catastrophic failures under adaptive attacks as shown in Section 3.1. Therefore, $\beta$ is typically recommended to be selected in the range $(0.5, 1)$ to reflect the reasonable assumption that out-links are more reliable than in-links but out-links are not fully trustworthy due to the existence of indirect in-link attacks on the neighbors.

**Advantages.** The proposed BBRW enjoys the advantages of simplicity, trustworthiness, explainability, universality, and efficiency. First, BBRW is simple due to its clear motivation and easy implementation. It is easy to tune with only one hyperparameter. Second, the hyperparameter $\beta$ provides the flexibility to adjust the trust between out-links and in-links, which helps avoid catastrophic failures and the false sense of robustness caused by the unconditional trust in out-links. The working mechanism and motivation of this hyperparameter are clearly justified by a theoretical analysis in Section 3.3. Moreover, it can be readily used as a plug-in layer to improve the robustness of various GNN backbones, and it shares the same computational and memory complexities as the backbone GNNs. BBRW is also compatible with existing defense strategies developed for undirected GNNs.

### 3.3 THEORETICAL ANALYSIS OF BBRW

We provide a theoretical analysis of BBRW to understand its working mechanism. Let $h_x^{(0)}$ and $h_x^{(k)}$ be the input feature and the $k$-th layer hidden feature of node $x$ in GNNs. The influence score $I(x,y) = \left\| \frac{\partial h_x^{(k)}}{\partial h_y^{(0)}} \right\|_1$ can measure the impact of node $y$ on node $x$ in the message passing (Xu et al., 2018). The attack mechanisms of out-link indirect attack and in-link direct attack are shown in Figure 3 (a). In our hyperparameter settings, we employ 2 layer neural networks for

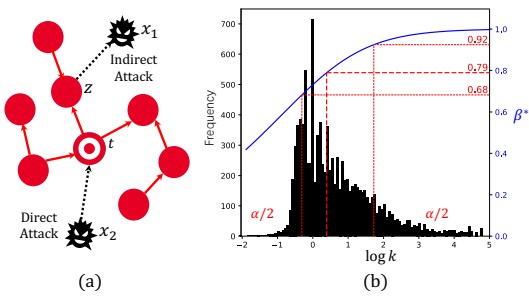

Figure 3: Theoretical analysis of BBRW.

BBRW, resulting in 2-step random walk message passing: $\tilde{\mathbf{A}}_\beta^2$. Therefore, the increment of influence score after conducting the attack on node $t$ is equivalent to $\tilde{\mathbf{A}}_\beta^2(t, x_1)$ for *indirect attack* or $\tilde{\mathbf{A}}_\beta^2(t, x_2)$ for *direct attack*. From the perspective of defense, we need to robustify the GNNs by choosing an appropriate $\beta$ to mitigate the potential impact of the stronger attack among them:

$$\beta^* = \arg \min_{\beta \in [0,1]} \max\{\tilde{\mathbf{A}}_\beta^2(t, x_1), \tilde{\mathbf{A}}_\beta^2(t, x_2)\}. \tag{1}$$

**Theorem 1.** *Define the degree difference factor as $k := \frac{\mathbf{D}_\beta^{-1}(t) + \mathbf{D}_\beta^{-1}(x_2)}{2\mathbf{D}_\beta^{-1}(z)}$, then the theoretical optimal $\beta^*$ in Eq. (1) is given by $\beta^*(k) = \sqrt{k^2 + 2k} - k$.*

For a target node $t$, Theorem 1 provides the theoretical optimal $\beta^*$ that minimizes the attack influence from a 1-hop direct attack or a 2-hop indirect attack through a neighbor node $z$. The detailed proof is presented in Appendix A.3. Since different nodes may have a different $\beta^*$ due to their distinct $k$, we perform a statistical analysis on the distribution of $\beta^*$. From Cora ML, we randomly select node $t$, $x_2$, and its neighbor $z$ such that we calculate and collect multiple samples $\{k^{(i)}\}_{i=1}^{10000}$. Figure 3 (b) shows the histogram of $\{\log k^{(i)}\}_{i=1}^{10000}$ and $\beta^*$ in terms of $\log k$. From the figure, we can find

that the optimal $\beta^*$ has a median of 0.79 and the $1 - \alpha$ ($\alpha = 0.2$) confidence interval is $(0.68, 0.92)$. This optimal value range aligns well with the optimally tuned $\beta^*$ in the ablation study in Section 4.3 (Figure 5 and Figure 6), further substantiating the validity of our approach.

## 4  EXPERIMENT

In this section, we provide comprehensive experiments to verify the advantages of the proposed BBRW. Comprehensive ablation studies are presented to illustrate the working mechanism of BBRW.

### 4.1  EXPERIMENTAL SETTING

**Datasets.** For the attack setting, we use the two most widely used datasets in the literature, namely Cora ML and Citeseer (Sen et al., 2008). We use the directed graphs downloaded from the work (Zhang et al., 2021) and follow their data splits (10% training, 10% validation, and 80% testing). We repeat the experiments for 10 random data splits and report the mean and variance of the node classification accuracy.

**Baselines.** We compare our models with seven undirected GNNs: GCN (Kipf & Welling, 2016), APPNP (Gasteiger et al., 2018), Jaccard-GCN (Wu et al., 2019), RGCN (Zhu et al., 2019), GRAND (Feng et al., 2020), GNNGuard (Zhang & Zitnik, 2020) and SoftMedian (Geisler et al., 2021), most of which are designed as robust GNNs. Additionally, we also select three state-of-the-art directed GNNs including DGCN (Tong et al., 2020b), DiGCN (Tong et al., 2020a) and MagNet (Zhang et al., 2021) as well as the graph-agnostic MLP.

**Hyperparameter settings.** For all methods, hyperparameters are tuned from the following search space: 1) learning rate: {0.05, 0.01, 0.005}; 2) weight decay: {5e-4, 5e-5, 5e-6}; 3) dropout rate: {0.0, 0.5, 0.8}. For APPNP, we use the teleport probability $\alpha = 0.1$ and propagation step $K = 10$ as (Gasteiger et al., 2018). For BBRW, we tune $\beta$ in $[0, 1]$ with the interval 0.1. For a fair comparison, the proposed BBRW-based methods share the same architectures and hyperparameters with the backbone models except for the plugged-in BBRW layer. For all models, we use 2 layer neural networks with 64 hidden units. Other hyperparameters follow the settings in their original papers.

**Adversary attacks & evaluations.** We conduct evasion target attacks using PGD topology attack (Xu et al., 2019) under the proposed RDGA setting. The details of the attacking algorithm are presented in Appendix A.1. We chose PGD attack because it is the strongest attack as verified by our experiments in Appendix A.2. We randomly select 20 target nodes per split for robustness evaluation and run the experiments for multiple link budgets $\Delta \in \{0\%, 25\%, 50\%, 100\%\}$ of the target node's total degree. *Transfer* and *Adaptive* refer to transfer and adaptive attacks, respectively. For transfer attacks, we choose a 2-layer GCN as the surrogate model following existing works (Mujkanovic et al., 2022; Zügner et al., 2018b). For adaptive attacks, the victim models are the same as the surrogate models, avoiding a false sense of robustness in transfer attacks. **In particular, the adaptive attack is executed after all the hyperparameters, including $\beta$, have been chosen for BBRW-based models.** "\" means we do not find a trivial solution for adaptive attack since it is non-trivial to compute the gradient of the adjacency matrix for those victim models.

### 4.2  ROBUST PERFORMANCE

To demonstrate the effectiveness, robustness, and universality of the proposed BBRW message-passing framework, we develop multiple variants of it by plugging BBRW into classic GNN backbones: GCN (Kipf & Welling, 2016), APPNP (Gasteiger et al., 2018) and SoftMedian (Geisler et al., 2021). The clean and robust performance are compared with plenty of representative GNN baselines on Cora-ML and Citeseer datasets as summarized in Table 2 and Table 3, respectively. From these results, we can observe the following:

- In most cases, all baseline GNNs underperform the graph-agnostic MLP under adaptive attacks, which indicates their incapability to robustly leverage graph topology information. However, most BBRW variants outperform MLP. Taking Cora-ML as an instance, the best BBRW variant (BBRW-SoftMedian) significantly outperforms MLP by $\{18\%, 16\%, 13.5\%\}$ (transfer attack) and $\{18.5\%, 14.5\%, 11\%\}$ (adaptive attack) under $\{25\%, 50\%, 100\%\}$ attack budgets. Even under 100% perturbation, BBRW-SoftMedian still achieves 84.5% robust accuracy under strong adaptive attacks, which suggests the value of trusting out-links.

- The proposed BBRW is a highly effective plug-in layer that significantly and consistently enhances the robustness of GNN backbones in both transfer and adaptive attack settings. Taking Cora-ML as an instance, under increasing attack budgets $\{25\%, 50\%, 100\%\}$: (1) BBRW-GCN outperforms GCN by $\{23.5\%, 45.5\%, 73\%\}$ (transfer attack) and $\{23\%, 44.5\%, 63\%\}$ (adaptive attack); (2) BBRW-APPNP outperforms APPNP by $\{7.5\%, 18.5\%, 39.5\%\}$ (transfer attack) and $\{7\%, 17\%, 25.5\%\}$ (adaptive attack); (3) BBRW-SoftMedian outperforms SoftMedian by $\{5.5\%, 14.5\%, 38.5\%\}$ (transfer attack) and $\{9\%, 15\%, 37\%\}$ (adaptive attack). The improvements are stronger under larger attack budgets.

- The proposed BBRW not only significantly outperforms existing directed GNNs such as DGCN, DiGCN, and MagNet in terms of robustness but also exhibits consistently better clean accuracy. BBRW also overwhelmingly outperforms existing robust GNNs under attacks. Compared with undirected GNN backbones such as GCN, APPNP, and SoftMedian, BBRW maintains the same or comparable clean accuracy.

Table 2: Classification accuracy (%) under different perturbation rates of graph attack. The best results are in **bold**, and the second-best results are underlined. (Cora-ML)

| Method | 0% | 25% | | 50% | | 100% | |
|---|---|---|---|---|---|---|---|
| | Clean | Transfer | Adaptive | Transfer | Adaptive | Transfer | Adaptive |
| MLP | 73.5±7.4 | 73.5±7.4 | 73.5±7.4 | 73.5±7.4 | 73.5±7.4 | 73.5±7.4 | 73.5±7.4 |
| DGCN | 89.5±7.6 | 76.5±13.0 | \ | 54.5±7.9 | \ | 38.0±14.2 | \ |
| DiGCN | 85.0±7.4 | 50.0±6.7 | \ | 40.5±9.1 | \ | 29.0±6.2 | \ |
| MagNet | 88.5±3.2 | 70.5±10.6 | \ | 59.5±10.6 | \ | 54.0±7.0 | \ |
| Jaccard-GCN | 90.5±6.5 | 69.5±7.9 | 65.5±7.9 | 44.0±6.2 | 34.0±7.0 | 21.0±7.0 | 8.0±4.6 |
| RGCN | 88.0±6.0 | 72.5±8.4 | 66.0±7.7 | 44.0±8.9 | 36.0±5.4 | 17.5±8.7 | 7.0±4.6 |
| GRAND | 85.5±6.1 | 74.0±7.0 | 65.0±7.4 | 64.0±9.2 | 51.0±8.6 | 45.0±7.1 | 24.0±7.7 |
| GNNGuard | 90.0±5.0 | 87.5±6.4 | 75.0±8.7 | 82.5±7.2 | 61.0±7.3 | 75.0±8.4 | 28.0±3.3 |
| GCN | 89.5±6.1 | 66.0±9.7 | 66.0±9.7 | 40.5±8.5 | 40.5±8.5 | 12.0±6.4 | 12.0±6.4 |
| BBRW-GCN | 90.0±5.5 | 89.5±6.1 | 89.0±6.2 | 86.0±5.4 | 85.0±6.3 | 85.0±7.1 | 75.0±10.2 |
| APPNP | 90.5±4.7 | 81.5±9.5 | 80.5±10.4 | 66.5±8.7 | 66.0±7.9 | 44.0±9.2 | 43.5±6.4 |
| BBRW-APPNP | 91.0±4.9 | 89.0±5.4 | 87.5±5.6 | 85.0±7.1 | 83.0±6.4 | 83.5±6.3 | 69.0±9.7 |
| SoftMedian | 91.5±5.5 | 86.0±7.0 | 83.0±7.1 | 75.0±8.4 | 73.0±7.1 | 48.5±11.4 | 47.5±9.3 |
| BBRW-SoftMedian | **92.0±4.6** | **91.5±5.0** | **92.0±4.6** | **89.5±6.9** | **88.0±5.1** | **87.0±8.4** | **84.5±8.8** |

## 4.3 ABLATION STUDY

In this section, we conduct further ablation studies on the attacking patterns, hyperparameter setting, and adversary capacity in RDGA to understand the working mechanism of the proposed BBRW.

**Attacking patterns.** In Table 2, we observe that BBRW-SoftMedian overwhelmingly outperform all baselines in terms of robustness. To investigate the reason, we show the adversarial attack patterns of transfer and adaptive attacks on BBRW-SoftMedian ($\beta = 0.7$) in Figure 4. In the transfer attack, the adversary spends 96.32% budget on in-links attacks on the target nodes directly, which causes a minor effect on BBRW-SoftMedian that trusts out-links more. In the adaptive attack, the adversary is aware of the biased trust of BBRW and realizes that in-links attacks are not sufficient. Therefore, besides direct in-link

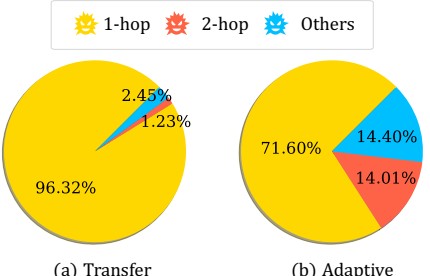

(a) Transfer   (b) Adaptive

Figure 4: Distributions of adversarial links.

attacks, it allocates 14.01% and 14.40% budgets to conduct the out-links indirect attacks on targets' neighbors and other attacks. Even though the adversary optimally adjusts the attack strategy, BBRW-SoftMedian still achieves an incredible 87% and 84.5% robust accuracy under 50% and 100% total attack budgets. This verifies BBRW's extraordinary capability to defend against adaptive attacks.

Table 3: Classification accuracy (%) under different perturbation rates of graph attack. The best results are in **bold**, and the second-best results are underlined. (Citeseer)

| Method | 0% | 25% | | 50% | | 100% | |
|---|---|---|---|---|---|---|---|
| | Clean | Transfer | Adaptive | Transfer | Adaptive | Transfer | Adaptive |
| MLP | 49.0±9.4 | 49.0±9.4 | 49.0±9.4 | 49.0±9.4 | **49.0±9.4** | **49.0±9.4** | **49.0±9.4** |
| DGCN | 64.0±7.0 | 54.0±8.3 | \ | 34.5±10.6 | \ | 27.0±10.1 | \ |
| DiGCN | 66.0±8.6 | 41.5±10.5 | \ | 29.5±8.2 | \ | 21.5±5.9 | \ |
| MagNet | 68.0±6.0 | 51.5±11.2 | \ | 35.0±12.0 | \ | 35.0±7.7 | \ |
| Jaccard-GCN | 57.0±7.1 | 45.5±7.9 | 38.5±9.5 | 23.0±7.8 | 11.5±5.5 | 20.0±10.2 | 6.5±5.0 |
| RGCN | 61.5±7.1 | 34.5±9.1 | 34.0±10.2 | 9.5±4.2 | 7.0±5.6 | 6.5±4.5 | 4.5±3.5 |
| GRAND | 67.5±6.0 | 56.5±6.3 | 56.0±8.9 | 43.0±5.1 | 42.5±9.0 | 37.5±8.1 | 27.5±6.8 |
| GNN-Guard | 60.5±7.2 | 50.0±8.7 | 43.5±9.0 | 33.0±8.7 | 18.0±8.4 | 31.5±8.7 | 8.5±3.9 |
| GCN | 59.0±5.4 | 36.5±9.5 | 36.5±9.5 | 10.5±5.7 | 10.5±5.7 | 4.5±4.2 | 4.5±4.2 |
| BBRW-GCN | 61.5±7.4 | 50.0±7.7 | 43.0±10.3 | 31.5±6.3 | 27.0±14.4 | 26.0±8.0 | 20.5±9.6 |
| APPNP | **72.0±6.0** | 53.5±9.5 | 51.0±6.2 | 16.0±10.7 | 13.5±98 | 9.0±4.4 | 8.5±9.0 |
| BBRW-APPNP | 69.0±4.4 | **66.0±8.3** | **59.0±9.7** | **55.0±8.1** | 26.5±8.4 | 43.5±6.3 | 14.5±6.1 |
| SoftMedian | 61.5±5.9 | 56.0±8.3 | 56.0±8.3 | 34.5±10.8 | 35.0±10.7 | 26.5±9.8 | 26.0±9.0 |
| BBRW-SoftMedian | 59.5±7.2 | 58.5±7.8 | 58.5±7.8 | 53.0±7.5 | 48.0±7.0 | **49.0±7.7** | 48.0±8.1 |

**Hyperparameter in BBRW.** BBRW is a simple and efficient approach. The only hyperparameter is the bias weight $\beta$ that provides the flexibility to differentiate and adjust the trust between out-links and in-links. We study the effect of $\beta$ by varying $\beta$ from 0 to 1 with an interval of 0.1 using BBRW-GCN. The accuracy under different attack budgets on Cora-ML is summarized in Figure 5. The accuracy on Citeseer is shown in Figure 6 in Appendix A.4. We can make the following observations:

- In terms of clean accuracy (0% attack budget), BBRW-GCN with $\beta$ ranging from 0.2 to 0.8 exhibit stable performance while the special case $\beta = 0$ and $\beta = 1$ (GCN-RW) perform worse. This suggests that both in-links and out-links provide useful graph information that is beneficial for clean performance, which is consistent with the conclusion in Section 3.1.

- Under transfer attacks, BBRW-GCN becomes more robust with the growth of $\beta$. It demonstrates that larger $\beta$ indeed can reduce the trust and impact of in-link attacks on target nodes.

- Under adaptive attacks, BBRW-GCN becomes more robust with the growth of $\beta$ but when it transits to the range close to $\beta = 1$ (GCN-RW), it suffers from catastrophic failures due to the indirect out-link attacks on targets' neighbors, which is consistent with the discovery in Section 3.1, This also explains the false sense of robustness evaluated under transfer attacks.

- The optimal values of $\beta$ align closely with our theoretical analysis in Section 3.3.

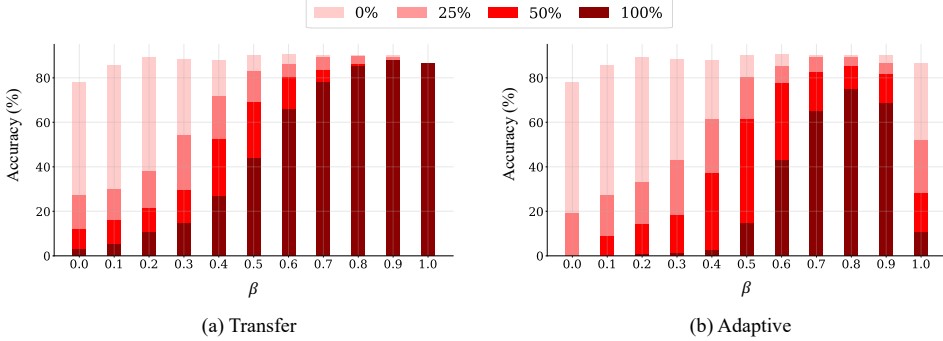

(a) Transfer      (b) Adaptive

Figure 5: Ablation study on $\beta$ (Cora-ML). Colors denote the accuracy under different attack budgets.

**Adversary capacity in RDGA.** One of the major reasons BBRW can achieve extraordinary robustness is to differentiate the roles and trust of in-links and out-links. In RDGA, we assume that the adversary can not manipulate the out-links of target nodes by fully masking target nodes' out-links (i.e., masking

rate=100%). This reflects the practical constraints in real-world applications as explained in Section 1 and Section 2. However, in reality, it is beneficial to consider more dangerous cases when the adversary may be able to manipulate some proportion of targets' out-links. Therefore, we also provide ablation study on the general RDGA setting by varying the masking rates of targets' out-links from 50% to 100%. The total attack budget including in-links and out-links is set as 50% of the degree of the target node. The results in Table 4 offer the following observations: (1) The robustness of undirected backbone GNNs is not affected by constraints on the out-link attacks of the target node, as they can't differentiate the out-links and in-links; (2) BBRW can significantly enhance the robustness of backbones models (e.g., SoftMedian) under varying masking rates. The improvements are stronger when out-links are better protected (higher mask rate).

Table 4: Ablation study on masking rates of target nodes' out-links under adaptive attack (Cora-ML).

| Model \ Masking Rate | 50% | 60% | 70% | 80% | 90% | 100% |
|---|---|---|---|---|---|---|
| GCN | 40.5±8.5 | 40.5±8.5 | 40.5±8.5 | 40.5±8.5 | 40.5±8.5 | 40.5±8.5 |
| BBRW-GCN | 52.0±11.4 | 54.5±10.8 | 56.5±9.2 | 60.0±10.4 | 60.5±11.0 | 85.0±6.3 |
| SoftMedian | 73.0±7.1 | 73.0±7.1 | 73.0±7.1 | 73.0±7.1 | 73.0±7.1 | 73.0±7.1 |
| BBRW-SoftMedian | 86.5±5.9 | 87.0±5.1 | 87.5±5.6 | 87.5±5.6 | 87.5±4.6 | 88.0±5.1 |

## 5 RELATED WORK

Existing research on the attacks and defenses of GNNs focuses on undirected GNNs that convert the graphs into undirected graphs (Chen et al., 2018; Zügner & Günnemann, 2019; Zügner et al., 2018b; Xu et al., 2019; Zhu et al., 2019; Zhang & Zitnik, 2020; Feng et al., 2020; Jin et al., 2020; Entezari et al., 2020; Geisler et al., 2021). Therefore, these works can not fully leverage the rich directed link information in directed graphs. A recent study (Mujkanovic et al., 2022) categorized 49 defenses published at major conferences/journals and evaluated 7 of them covering the spectrum of all defense techniques under adaptive attacks. Their systematic evaluations show that while some defenses are effective, their robustness is much lower than claimed in their original papers under stronger adaptive attacks. This not only reveals the pitfall of the false sense of robustness but also calls for new effective solutions. Our work differs from existing works by studying robust GNNs in the context of directed graphs, which provides unprecedented opportunities for improvements orthogonal to existing efforts.

There exist multiple directed GNNs designed for directed graphs but the robustness is largely unexplored. The work (Ma et al., 2019) proposes a spectral-based GCN for directed graphs by constructing a directed Laplacian matrix using the random walk matrix and its stationary distribution. DGCN (Tong et al., 2020b) extends spectral-based graph convolution to directed graphs by utilizing first-order and second-order proximity. MotifNet (Monti et al., 2018) uses convolution-like anisotropic graph filters based on local sub-graph structures called motifs. DiGCN (Tong et al., 2020a) proposed a directed Laplacian matrix based on the PageRank matrix. MagNet (Zhang et al., 2021) utilizes a complex Hermitian matrix called the magnetic Laplacian to encode undirected geometric structures in the magnitudes and directional information in the phases. GNNGuard (Zhang & Zitnik, 2020) introduces a robust propagation through reweighting and can be potentially extended to direct graphs, but it does not leverage the directional information to enhance robustness. The BBRW proposed in this work is a general framework that can equip various GNNs with the superior capability to handle directed graphs more effectively.

## 6 CONCLUSION

This work conducts a novel exploration of the robustness and trustworthiness of GNNs in the context of directed graphs. To achieve this objective, we introduce a new and more realistic graph attack setting for directed graphs. Additionally, we propose a simple and effective message-passing approach as a plug-in layer to significantly enhance the robustness of various GNN backbones, tremendously surpassing the performance of existing methods. Although the primary focus of this study is evasion targeted attack, the valuable findings reveal the substantial potential of leveraging the directional information in directed graphs to enhance the robustness of GNNs. Moving forward, further exploration of this potential will encompass various attack settings such as poison attacks and global attacks.

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
