# A  APPENDIX

In this appendix, we provide details about the theoretical analysis and additional experimental results that can not be fitted into the main paper due to space limitations.

## A.1  DETAILS OF PGD UNDER RDGA SETTING

The proposed Restricted Direct Graph Attack (RDGA) setting provides a more realistic attack budget allocation that differentiates out-link and in-link attacks. In principle, it is compatible with any existing graph attack algorithms such as PGD Xu et al. (2019) and Nettack Zügner et al. (2018a) by adjusting the attack budgets for out-links and in-links when selecting the edges. Due to the excellent attack performance of PGD Xu et al. (2019), we mainly adopt PGD as the attacking algorithm in this work. Specifically, we use the masking matrix $\mathbf{M}$ as described in Section 2.2 to zero out the gradients of the out-links of the target nodes during gradient descent iterations. The details of the attacking process are summarized in Algorithm 1.

---

**Algorithm 1** PGD attack under RDGA setting

---

**Input:** initial perturbation $\mathbf{P}^{(0)}$, budget $\Delta$, learning rate $\eta$, iterations $T$, number of random trials $K$
**Output:** optimal perturbation $\mathbf{P}^*$
1: **for** $t = 1, 2, ..., T$ **do**
2:     Restricted gradient descent: $\mathbf{P}^{(t)} = \mathbf{P}^{(t-1)} - \eta \nabla \ell(\mathbf{P}^{(t-1)}) \odot \mathbf{M}$
3: **for** $k = 1, 2, ..., K$ **do**
4:     Draw binary matrix $\mathbf{S}^{(k)}$ following

$$\mathbf{S}_{ij}^{(k)} = \begin{cases} 1 & \text{with probability } \mathbf{P}_{ij}^{(T)} \\ 0 & \text{with probability } 1 - \mathbf{P}_{ij}^{(T)} \end{cases}$$

5: Choose a perturbation $\mathbf{P}^*$ from $\{\mathbf{S}^{(k)}\}$ which yields the smallest attack loss $\ell(\mathbf{S}^{(k)})$ under $\|\mathbf{P}^*\|_0 \le \Delta$

---

## A.2  DIFFERENT ATTACKS UNDER RDGA SETTING

In Section 2.2, we mention that the greedy approaches such as FGA Mujkanovic et al. (2022) and Nettack (Zügner et al., 2018b) tend to be relatively weaker and we adopt PGD (Xu et al., 2019) under RDGA setting in our work. Here, we also provide the experimental results of FGA Mujkanovic et al. (2022) and Nettack (Zügner et al., 2018b) under RDGA setting to validate the aforementioned statement. We present the results on Coral-ML of 50% and 100% budgets in Table 5 and Table 6.

Table 5: Classification accuracy (%) under different attacks (Cora-ML, budget=50%)

| Attack \ Model | BBRW-GCN | BBRW-APPNP | BBRW-SoftMedian |
|---|---|---|---|
| RDGA-FGA | 84.5±6.5 | 85.0±7.1 | 91.5±5.9 |
| RDGA-Nettack | 88.0±6.0 | 84.5±6.1 | 89.0±5.4 |
| RDGA-PGD | 85.0±6.3 | 83.0±6.4 | 88.0±5.1 |

Table 6: Classification accuracy (%) under different attacks (Cora-ML, budget=100%)

| Attack \ Model | BBRW-GCN | BBRW-APPNP | BBRW-SoftMedian |
|---|---|---|---|
| RDGA-FGA | 77.0±10.1 | 77.0±6.4 | 92.0±6.4 |
| RDGA-Nettack | 85.5±7.2 | 79.0±6.6 | 86.5±6.7 |
| RDGA-PGD | 75.0±10.2 | 69.0±9.7 | 84.5±8.8 |

### A.3 PROOF OF THEOREM

**Theorem 1.** *Define the degree difference factor as $k := \frac{\mathbf{D}_\beta^{-1}(t) + \mathbf{D}_\beta^{-1}(x_2)}{2\mathbf{D}_\beta^{-1}(z)}$, then the theoretical optimal $\beta^*$ in Eq. (1) is given by $\beta^*(k) = \sqrt{k^2 + 2k} - k$.*

*Proof.* The increment of influence score $\Delta I$ after conducting the indirect attack on node $t$ is equivalent to:

$$\tilde{\mathbf{A}}_\beta^2(t, x_1) = \tilde{\mathbf{A}}_\beta(t, z) \cdot \tilde{\mathbf{A}}_\beta(z, x_1) = \frac{\beta}{\mathbf{D}_\beta(t)} \cdot \frac{\beta}{\mathbf{D}_\beta(z)},$$

and the increment of influence score after applying a direct attack on node $t$ will be proportional to:

$$\tilde{\mathbf{A}}_\beta^2(t, x_2) = \tilde{\mathbf{A}}_\beta(t, t) \cdot \tilde{\mathbf{A}}_\beta(t, x_2) + \tilde{\mathbf{A}}_\beta(t, x_2) \cdot \tilde{\mathbf{A}}_\beta(x_2, x_2) = \frac{1}{\mathbf{D}_\beta(t)} \cdot \frac{1 - \beta}{\mathbf{D}_\beta(t)} + \frac{1 - \beta}{\mathbf{D}_\beta(t)} \cdot \frac{1}{\mathbf{D}_\beta(x_2)}.$$

According to Eq. (1) and the definition of $k$, we have:

$$\begin{aligned}
\beta^* &= \arg\min_{\beta \in [0,1]} \max\{\tilde{\mathbf{A}}_\beta^2(t, x_1), \tilde{\mathbf{A}}_\beta^2(t, x_2)\} \\
&= \arg\min_{\beta \in [0,1]} \max\{\frac{\beta}{\mathbf{D}_\beta(t)} \cdot \frac{\beta}{\mathbf{D}_\beta(z)}, \frac{1}{\mathbf{D}_\beta(t)} \cdot \frac{1 - \beta}{\mathbf{D}_\beta(t)} + \frac{1 - \beta}{\mathbf{D}_\beta(t)} \cdot \frac{1}{\mathbf{D}_\beta(x_2)}\} \\
&= \arg\min_{\beta \in [0,1]} \max\{\beta^2 \cdot \mathbf{D}_\beta^{-1}(z), (1 - \beta) \cdot [\mathbf{D}_\beta^{-1}(t) + \mathbf{D}_\beta^{-1}(x_2)]\} \\
&= \arg\min_{\beta \in [0,1]} \max\{\beta^2, k(1 - \beta)\} \\
&= \sqrt{k^2 + 2k} - k.
\end{aligned}$$

$\square$

## A.4 SUPPLEMENTARY EXPERIMENTS

Figure 6 presents the ablation study on hyperparameter $\beta$ on Citeseer dataset. Figure 7 presents the distribution of adversarial links under different attack budgets, indicating the adversary's distinct attacking behaviors when attacking different victim models adaptively.

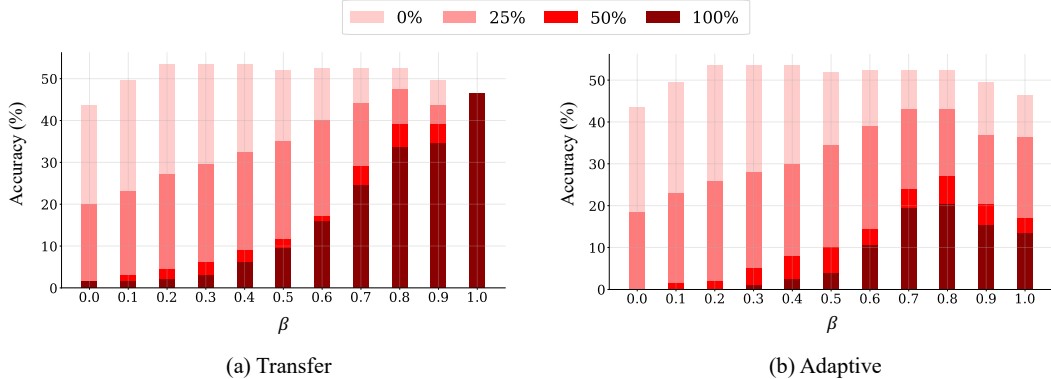

(a) Transfer  (b) Adaptive

Figure 6: Ablation study on $\beta$ (Citeseer). Colors denote the accuracy under different attack budgets.

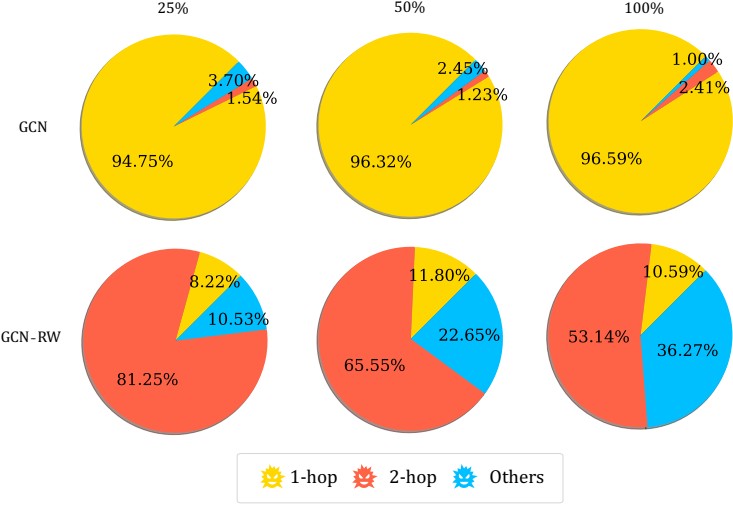

Figure 7: Distributions of adversarial links against different victim models (GCN and GCN-RW) under different attack budgets (25%, 50%, 100%). The yellow portion represents attacks by 1-hop neighbors on the target (direct in-link attacks); the red portion represents attacks by 2-hop neighbors on the targets' neighbors (indirect out-link attacks); and the blue portion represents other attacks.