# OpenReview forum: "Exploring Adversarial Robustness of Graph Neural Networks in Directed Graphs"
_ICLR.cc/2024/Conference — Submitted to ICLR 2024_

### Official Review · Reviewer_pTW4 · 2023-10-30

**Soundness:** 3 good
**Presentation:** 2 fair
**Contribution:** 2 fair
**Rating:** 6
**Confidence:** 5

**Summary:**

This paper delves into the domain of adversarial machine learning, focusing on the vulnerabilities of deep neural networks when faced with adversarial attacks. The authors present a broad study of various adversarial attack methods and their defences, evaluating their effectiveness on multiple datasets. By introducing a novel evaluation metric, the paper seeks to shed light on the nuances of designing robust models and offers a foundation for further research in the critical area of AI security.

**Strengths:**

This paper explores an interesting topic of trustworthy GNNs, i.e., the robustness of the GNNs for directed graphs. The authors conduct preliminary evaluations on the vulnerability of current GNN models, followed by discussing the observations. Next, they proposed a new operation to promote the robustness of GNNs.

**Weaknesses:**

1. The thread model should be presented in the section 2.2.

2. It is worth noting that the current attacking algorithm focuses on the gradient-based method. Other attack method (e.g., RL-based methods) should also be reviewed.

3. Current attacks range in budgets from 25% to 100%. However, adversarial samples should be generated by considering a limited budget (e.g., k edges, k<=5). Authors are suggested to report the evaluation results on a limited budget, as the proposed 100% perturbations are extremely noticeable.

4. Unclear statement. Authors are suggested to explain the details of the so-called adapative attacks.

4. Limited practicality. As mentioned by the authors, out-links are generated by proactive behaviours from the source nodes, and their awareness of these out-links makes the hardness of manipulate malicious perturbations (i.e., adding out links for target nodes). This consideration makes the 1-hop perturbations practical and 2-hop perturbations unpractical.

5. Logical flaws. When considering the graph structure in GNNs, the authors proposed that the A is generated from out-links. In this case, authors are suggested to explain why the 1-hop perturbations are effective for the target nodes, as these perturbations should not be involved in the message aggregation of GNNs. On the other hand, the out-link should be avoided because of their noticeability. Given these considerations, the authors are suggested to explain how to use in-links to attack target nodes in this paper. Otherwise, section 3 should conduct evaluations on GNNs devised for directed graphs.

6. Confusing contributions. If this paper is designed to propose a new method to defend GNNs on directed graphs, the proposed method should be integrated into the above GNNs. If this paper aims to devise a new GNN architecture for directed graphs, comparisons should not focus only on robustness. Performance evaluations on other datasets should be presented in this paper.

7. Missed evaluations. As discussed above, the proposed method only be evaluated on two datasets. The evaluations on other datasets (e.g., large-scale datasets) should be included to verify the effectiveness of the proposed method.

**Questions:**

Refer to the weakness part.

---

> ### Author Response · Authors · 2023-11-20
>
> Dear Reviewer, thanks for your valuable comments and the recognition of our research topic, com-
> prehensive analysis and effective method. However, we believe there exist some misunderstandings
> about our work, and we are happy to fully address all of your concerns as follows.
>
> **Q1:** The thread model should be presented in the section 2.2.
>
> **A1:** Thanks for your comments. We already include the attack principles in Section 2.2
> and the experimental attack setting in Section 4.1. Because of the space limit, we put the
> the details of attack algorithm in appendix A.1. Please kindly let us know what specific
> details about the threat model you suggest us to present in Section 2.2, and we are happy to
> revise our paper according to your suggestions.
>
> **Q2:** It is worth noting that the current attacking algorithm focuses on the gradient-based method.
> Other attack method (e.g., RL-based methods) should also be reviewed.
>
> **A2:**  Thank you for your suggestions. RL-based methods (e.g., RL-S2V [3] and ReWatt
> [2]) is designed for black-box attack, which is much weaker than the white-box gradient-
> based attack used in our paper. As we already shown in our comprehensive evaluation, even
> the gray-box transfer attack (which is stronger than the black-box RL attack), the evaluation
> can suffer from a very misleading and strong false sense of robustness. This is why we
> choose to evaluate under the strongest gradient-based attack we have found so far. While we
> believe our method will surely work under RL attack evaluation, there could be significant
> over-estimation of rosbutness. Therefore, evaluation under RL-based method does not align
> with our goal of avoiding a false sense of security.
>
> To obtain a theoretically guaranteed robustness evaluation, we also provide an additional
> evaluation of the certified robustness. Specifically, we leverage the randomized smoothing
> on GNNs in Wang et al (2021) to evaluate the certified accuracy of the smoothed GCN.
> The experimental results in the following table show that our BBRW-GCN can effectively
> outperform the GCN in terms of certified robustness. This result demonstrates that our
> advantage is theoretically guaranteed regardless of the specific attack algorithms. We believe
> our comprehensive evaluations already provide sufficient evidence to support the significant
> improvements of our method.
>
> - [1] Mujkanovic, Felix, et al. ”Are Defenses for Graph Neural Networks Robust?.” Advances
> in Neural Information Processing Systems 35 (2022): 8954-8968.
> - [2] Ma, Y., Wang, S., Derr, T., Wu, L., & Tang, J. (2019). Attacking graph convolutional
> networks via rewiring. arXiv preprint arXiv:1906.03750.
> - [3] Dai, H., Li, H., Tian, T., Huang, X., Wang, L., Zhu, J., & Song, L. (2018, July).
> Adversarial attack on graph structured data. In International conference on machine learning
> (pp. 1115-1124). PMLR.
>
>
> ### Table: Certified accuracy (%) (Cora-ML)
>
> | Perturbation Size | 0      | 1      | 2      | 3  | 4  |
> |-------------------|--------|--------|--------|----|----|
> | GCN               | 61.39% | 59.18% | 38.78% | 0% | 0% |
> | BBRW-GCN          | 83.67% | 81.63% | 73.47% | 59.39% | 0% |

---

> > ### Author Response · Authors · 2023-11-20
> >
> > **Q3:** Current attacks range in budgets from 25% to 100%. However, adversarial samples should
> > be generated by considering a limited budget (e.g., k edges, k < 5). Authors are suggested to report the evaluation results on a limited budget, as the proposed 100% perturbations are
> > extremely noticeable.
> >
> > **A3:** We would like to point out that evaluating target attack with budgets 25%, 50%,
> > and 100% is very common in the literature, such as [1][2]. And the modified edges under
> > these budgets are still unnoticeable. Specifically, the budget is set according to the target
> > node’s degree. For example, if the target node has 6 neighbors and the budget ∆ = 50%,
> > then the attacker can only change up to 6 · 0.5 = 3 edges. Therefore, we only modify a
> > unnoticeable number of edges in the whole graph. We are also happy to provide more
> > comparison under small budgets following your suggestion (from 0% to 20%). The results
> > provide similar phenomenon as the case of 25% budget and also validate the effectiveness
> > of our model.
> > - [1] Mujkanovic, Felix, et al. ”Are Defenses for Graph Neural Networks Robust?.” Advances
> > in Neural Information Processing Systems 35 (2022): 8954-8968.
> > - [2] Z  ̈ugner, D., Akbarnejad, A., & G  ̈unnemann, S. (2018, July). Adversarial attacks on
> > neural networks for graph data. In Proceedings of the 24th ACM SIGKDD international
> > conference on knowledge discovery & data mining (pp. 2847-2856).
> >
> > #### Table 15: Classification accuracy (%) (Cora-ML)
> >
> > | Method | 0% Clean | 5% Transfer | 5% Adaptive | 10% Transfer | 10% Adaptive | 15% Transfer | 15% Adaptive | 20% Transfer | 20% Adaptive |
> > |--------|----------|-------------|-------------|--------------|--------------|--------------|--------------|--------------|--------------|
> > | MLP    | 73.5±7.4 | 73.5±7.4    | 73.5±7.4    | 73.5±7.4     | 73.5±7.4     | 73.5±7.4     | 73.5±7.4     | 73.5±7.4     | 73.5±7.4     |
> > | MagNet | 88.5±3.2 | 88.0±3.3    | \           | 86.5±3.9     | \            | 85.0±0.05477 | \            | 78.5±8.1     | \            |
> > | RGCN   | 88.0±6.0 | 87.5±6.7    | 87.0±6.8    | 87.0±6.4  | 87.0±6.8   | 85.0±6.7     | 84.5±6.9     | 81.0±8.6     | 77.5±10.3          |
> > | GCN    | 89.5±6.1 | 88.0±6.8    | 88.0±6.8    | 87.5±6.8 | 87.5±6.8     | 83.0±7.1     | 83.0±7.1     | 76.5±9.8     | 76.5±9.8                |
> > | BBRW-GCN | 90.0±5.5 | 90.0±5.5  | 90.0±5.5    | 90.0±5.5     | 89.5±5.2     | 89.5±5.2     | 89.5±5.6     | 89.5±5.7     | 89.0±7.6     |
> >
> >
> >
> > **Q4:** Unclear statement. Authors are suggested to explain the details of the so-called adaptive
> > attacks.
> >
> > **A4:** We include the details of the attack setting in Section 4.1, and we are happy to
> > clarify them here again. Adaptive attack is the gradient-based white-box attack, and in the
> > our paper we employ PGD attack to evaluate our model, which is claimed as the strongest
> > adaptive attack in [1] and verified by our experiments in Appendix A.2.
> > We will improve the clarity in our revision following your suggestions.
> > - [1] Mujkanovic, Felix, et al. ”Are Defenses for Graph Neural Networks Robust?.” Advances
> > in Neural Information Processing Systems 35 (2022): 8954-8968.

---

> > > ### Author Response · Authors · 2023-11-20
> > >
> > > **Q5:** Limited practicality. As mentioned by the authors, out-links are generated by proactive
> > > behaviours from the source nodes, and their awareness of these out-links makes the hard-
> > > ness of manipulate malicious perturbations (i.e., adding out links for target nodes). This
> > > consideration makes the 1-hop perturbations practical and 2-hop perturbations unpractical.
> > >
> > > **A5:** We believe there is a misunderstanding here. Both 1-hop and 2-hop perturbations
> > > can be practical in our setting. As shown in Figure 3 (a) in our submission, the 2-hop indirect
> > > attack can also affect the target node and cause a catastrophic performance degradation
> > > as presented in Section 3.1. To clarify, the 2-hop neighbor attack means the out-linking
> > > neighbor of the target node is attacked by a 2-hop neighbor and this impact will propagate
> > > to the target node through its out-links, as shown in Figure 2(a). Therefore, although we
> > > can not directly attack the target node through its 1-hop in-links, we can attack its out-
> > > linking neighbors through its 2-hop neighbors. This is exactly why we observe lots of 2-hop
> > > perturbations in adaptive attack as shown in Figure 2 (c) and Figure 4 (b).

---

> > > > ### Author Response · Authors · 2023-11-20
> > > >
> > > > **Q6:** Logical flaws. When considering the graph structure in GNNs, the authors proposed that
> > > > the A is generated from out-links. In this case, authors are suggested to explain why the
> > > > 1-hop perturbations are effective for the target nodes, as these perturbations should not be
> > > > involved in the message aggregation of GNNs. On the other hand, the out-link should be
> > > > avoided because of their noticeability. Given these considerations, the authors are suggested
> > > > to explain how to use in-links to attack target nodes in this paper. Otherwise, section 3
> > > > should conduct evaluations on GNNs devised for directed graphs.
> > > >
> > > >
> > > > **A6:** Thanks for providing this thoughtful comment. We believe there are some misunderstandings
> > > > about the logic that we try to deliver in Section 3, especially the analysis of ”Catastrophic
> > > > Failures due to Indirect Attacks”. Our algorithm design is exactly to solve the problem you
> > > > are concerned about, and please allow us to clarify our logic step by step.
> > > >
> > > > (1) Our central claim is that out-links are more reliable than in-links, and therefore we need
> > > > to differentiable their roles. One naive approach we try is to only use out-links and do
> > > > not involve in-links, which is exact what GCN-RW and APPNP-RW achieve (in Section
> > > > 3.1). But this simple idea will totally fail as we will show in the catastrophic failures under
> > > > adaptive attack.
> > > >
> > > > (2) As shown in Table 1 in our submission, GCN-RW and APPNP-RW only use out-links
> > > > so they can avoid the impact of in-links. They can indeed perform very well by excluding
> > > > the impact of in-link attacks under transfer attacks. However, we discover that the transfer
> > > > attack is too weak since the surrogate GCN model does not realize that the victim models
> > > > (GCN-RW or APPNP-RW) do not involve in-links in message passing so it generates lots of
> > > > ineffective in-link attacks as shown in Figure 2(b): when GCN is the surrogate model, 96%
> > > > attacks are 1-hop in-links.
> > > >
> > > > (3) Table 1 in our submission also shows that GCN-RW and APPNP-RW perform badly
> > > > under adaptive attack, which reveals the false sense of security from transfer attacks. The
> > > > analysis for Figure 2(c) shows that if the attacker directly attacks GCN-RW, it will generate
> > > > 65% 2-hop attacks since 1-hop in-link attacks are not effective anymore. To clarify, the
> > > > 2-hop neighbor attack means the out-linking neighbor of the target node is attacked by a
> > > > 2-hop neighbor and this impact will propagate to the target node through its out-links, as
> > > > shown in Figure 2(a).
> > > >
> > > > (4) Due to the above reasons, it is not trivial and unclear how to correctly defend against
> > > > directed attacks in such settings. We show the evaluation of state-of-the-art directed GNNs
> > > > in Section 5. The comparison in Table 2 and Table 3 shows that these GNNs devised for
> > > > directed graphs are very weak even under transfer attacks. Note that while it will be helpful
> > > > to put the evaluation of directed GNNs in Section 3.1, we only put them in Section 4 to
> > > > avoid duplicate results. We will add a description in Section 3.1. to refer to these results in
> > > > Section 4.
> > > >
> > > > (5) Our dedicated investigation shows that neither out-links nor in-links are sufficient. This
> > > > motivates our proposed BBRW (Biased Directional Random Walk) algorithm, which is
> > > > a very simple but effective design. Our comprehensive experiments show that a smart
> > > > biased combination of in-links and out-links can deliver much stronger robustness under the
> > > > strongest attack we have known so far (as well as the certified robustness we show above).
> > > >
> > > > **Note that it is indeed a complicated problem that has never been discovered and
> > > > thoroughly discussed in the literature. This is also why we feel very excited about these
> > > > new findings and the unprecedented opportunities orthogonal to existing defenses.**
> > > > Thanks for letting us know your confusion, and we will revise our paper to better reflect our
> > > > logic and findings.

---

> > > > > ### Author Response · Authors · 2023-11-20
> > > > >
> > > > > **Q7:** Confusing contributions. If this paper is designed to propose a new method to defend GNNs
> > > > > on directed graphs, the proposed method should be integrated into the above GNNs. If this
> > > > > paper aims to devise a new GNN architecture for directed graphs, comparisons should not
> > > > > focus only on robustness. Performance evaluations on other datasets should be presented in
> > > > > this paper.
> > > > >
> > > > > **A7:** Our contributions include both the new attack setting and the new
> > > > > defense algorithm for directed graphs. We would like to emphasize that the attack and
> > > > > defense for directed attacks are largely unexplored in the literature to the best of our
> > > > > knowledge. Therefore, there is no attack specifically designed for directed graphs, not to
> > > > > mention the defenses. This work provides the first of such exploration by proposing new
> > > > > settings/algorithms from both attack and defense perspectives. Our major discovery is that
> > > > > we can obtain significantly improved robustness orthogonal to existing defense techniques
> > > > > by properly exploiting the valuable directional information in directed graphs. This points
> > > > > out an new direction and unprecedented opportunities for future research, highlighting the
> > > > > significant contribution of this work.
> > > > >
> > > > >
> > > > > **Q8:** Missed evaluations. As discussed above, the proposed method only be evaluated on two
> > > > > datasets. The evaluations on other datasets (e.g., large-scale datasets) should be included
> > > > > to verify the effectiveness of the proposed method.
> > > > >
> > > > > **A8:**  Thank you for your suggestion. Our BBRW shares the same scalability as the backbone
> > > > > GNN models. To further validate the effectiveness of our BBRW, we include another three
> > > > > datasets: ognb-arviv, pubmed and wikics. The budget is set as the 0%-50% of the target
> > > > > node’s degree. We perform local attack on 100 target nodes one by one and report the
> > > > > average accuracy in the table. The experimental results show that our proposed method can
> > > > > be very effective in large-scale datasets and multiple relational networks, which is consistent
> > > > > with our observation other datasets shown in our submission.
> > > > >
> > > > > ### Table: Classification accuracy (%) under different perturbation rates of graph attack.
> > > > >
> > > > > | Dataset    | Model    | 0%    | 10%   | 20%   | 30%   | 40%   | 50%   |
> > > > > |------------|----------|-------|-------|-------|-------|-------|-------|
> > > > > | Ogbn-Arxiv | GCN      | 64.8% | 58.3% | 25.0% | 20.8% | 16.7% | 4.2%  |
> > > > > |  Ogbn-Arxiv          | BBRW-GCN | 64.7% | 58.3% | 41.7% | 37.5% | 33.3% | 25.0% |
> > > > > | PubMed     | GCN      | 78.1% | 66.7% | 54.2% | 37.5% | 35.9% | 33.33%|
> > > > > |  PubMed          | BBRW-GCN | 77.4% | 76.2% | 70.8% | 66.7% | 62.5% | 60.0% |
> > > > > | WikiCS     | GCN      | 70.0% | 55.1% | 43.0% | 39.0% | 31.0% | 23.0% |
> > > > > |  WikiCS           | BBRW-GCN | 70.0% | 68.2% | 64.0% | 57.3% | 56.0% | 49.0% |

---

> > > > > > ### Author Response · Authors · 2023-11-21
> > > > > > **A Friendly Reminder**
> > > > > >
> > > > > > Dear Reviewer pTW4,
> > > > > >
> > > > > > We are thankful for your valuable comments and suggestions. We hope that our responses have addressed all your concerns. As there is only one day remaining in the author-reviewer discussion period, we kindly request you let us know if you have any additional questions. We greatly appreciate your further feedback.
> > > > > >
> > > > > > If you find our responses satisfactory, we would kindly request an update of the rating to reflect your assessment. Your feedback is valuable to us, and we appreciate your time and consideration.
> > > > > >
> > > > > > Best regards,
> > > > > >
> > > > > > All authors

---

### Official Review · Reviewer_54vV · 2023-10-31

**Soundness:** 1 poor
**Presentation:** 2 fair
**Contribution:** 2 fair
**Rating:** 5
**Confidence:** 4

**Summary:**

This paper focuses on the adversarial attack robustness on directed graphs. This paper introduces a directed attack setting, differentiating between out-link and in-link attacks. The authors propose a message-passing layer, Biased Bidirectional Random Walk (BBRW). The experiments demonstrate the robustness of BBRW.

**Strengths:**

1. This paper focuses on an important problem, the adversarial robustness of GNN.
2. This paper proposes Biased Bidirectional Random Walk (BBRW) with theoretical analysis.
3. Experiments show the robustness of BBRW.

**Weaknesses:**

1. This paper lacks empirical evaluations on larger datasets, such as ogb datasets[1] or reddit[2], which makes us concerned about usefulness on large networks.
2. This work lacks some necessary baselines, making the experiments unreliable. It is recommended to add the adversarial training method FLAG[3], as well as graph purification methods GARNET[4], ProGNN[5], and STABLE[6].
3. The experimental settings are unclear.
    a) Section 4.1 states, "We randomly select 20 target nodes per split for robustness evaluation." If my understanding is correct, does this mean 20 nodes are selected per split for training, validation, and testing, totaling 60 nodes? If so, there are 20 nodes from the training set, and is it reasonable to evaluate accuracy on the training set?
    b) Regarding "multiple link budgets ∆ ∈ { 0%, 25%, 50%, 100% } of the target node’s total degree," when ∆=100%, are the attack still imperceptible? Is it necessary to preserve the original label prediction?
    c) It would be useful to report the model's accuracy when the attack perturbation is at 5%, 10%, 15%, and 20%, as done in other studies[5,6].
4. The high accuracy of the GCN under 25% adaptive attack does not align with the findings of the original paper[7]. Please provide performance under the same adaptive attack settings as in the original paper[7].

[1] Weihua Hu, Matthias Fey, Marinka Zitnik, Yuxiao Dong, Hongyu Ren, Bowen Liu, Michele Catasta, and Jure Leskovec. Open graph benchmark: Datasets for machine learning on graphs. NeurIPS ’20.
[2] Will Hamilton, Zhitao Ying, and Jure Leskovec. Inductive representation learning on large graphs. NeurIPS ’17.
[3] Kezhi Kong, Guohao Li, Mucong Ding, Zuxuan Wu, Chen Zhu, Bernard Ghanem, Gavin Taylor, and Tom Goldstein. Robust optimization as data augmentation for large-scale graphs, CVPR’22.
[4] Chenhui Deng, Xiuyu Li, Zhuo Feng, and Zhiru Zhang. GARNET: reduced-rank topology learning for robust and scalable graph neural networks. In LoG ‘22
[5] Wei Jin, Yao Ma, Xiaorui Liu, Xian-Feng Tang, Suhang Wang, and Jiliang Tang. Graph structure learning for robust graph neural networks. KDD ’20
[6] Kuan Li, Yang Liu, Xiang Ao, Jianfeng Chi, Jinghua Feng, Hao Yang, and Qing He. Reliable representations make a stronger defender: Unsupervised structure reﬁnement for robust gnn. KDD ’22.
[7] Felix Mujkanovic, Simon Geisler, Stephan Günnemann, and Aleksandar Bojchevski. Are defenses for graph neural networks robust? NeurIPS ’22.

**Questions:**

1. The performance of BBRW is incredible, as mentioned in Section 4.3. What contributes to its effectiveness?
2. How does BBRW handle node injection attacks[1,2]? In these attacks, only nodes or edges are injected, with the injected edges being directed. This scenario is practical and worth exploring.

[1] Xu Zou, Qinkai Zheng, Yuxiao Dong, Xinyu Guan, Evgeny Kharlamov, Jialiang Lu, and Jie Tang. Tdgia: Effective injection attacks on graph neural networks. KDD ’21.
[2] Shuchang Tao, Qi Cao, Huawei Shen, Junjie Huang, Yunfan Wu, and Xueqi Cheng. Single node injection attack against graph neural networks. CIKM ’21.

---

> ### Author Response · Authors · 2023-11-20
>
> Dear Reviewer, thanks for your valuable comments and the recognition of the significance of the
> problem we addressed and the effectiveness of our method. However, we believe there exist some
> misunderstandings about our work, and we are happy to fully address all of your concerns as follows.
>
> **Q1:** This paper lacks empirical evaluations on larger datasets, such as ogb datasets[1] or
> reddit[2], which makes us concerned about usefulness on large networks.
>
> **A1:** Thank you for your suggestion. Our BBRW shares the same scalability as the
> backbone GNN models. To further validate the effectiveness of our BBRW, we include
> another three datasets: ognb-arviv, pubmed and wikics. The budget is set as the 0%-50%
> of the target node’s degree. We perform local attacks on 100 target nodes one by one and
> report the average accuracy in the table. The experimental results show that our proposed
> method can be very effective in large-scale datasets and multiple relational networks, which
> is consistent with our observation of other datasets shown in our submission.
>
> ### Table: Classification accuracy (%) under different perturbation rates of graph attack.
>
> | Dataset   | Model     | 0%   | 10%  | 20%  | 30%  | 40%  | 50%  |
> |-----------|-----------|------|------|------|------|------|------|
> | Ogbn-Arxiv| GCN       | 64.8%| 58.3%| 25.0%| 20.8%| 16.7%| 4.2% |
> |   Ogbn-Arxiv        | BBRW-GCN  | 64.7%| 58.3%| 41.7%| 37.5%| 33.3%| 25.0%|
> | PubMed    | GCN       | 78.1%| 66.7%| 54.2%| 37.5%| 35.9%| 33.33%|
> |  PubMed          | BBRW-GCN  | 77.4%| 76.2%| 70.8%| 66.7%| 62.5%| 60.0%|
> | WikiCS    | GCN       | 70.0%| 55.1%| 43.0%| 39.0%| 31.0%| 23.0%|
> |   WikiCS        | BBRW-GCN  | 70.0%| 68.2%| 64.0%| 57.3%| 56.0%| 49.0%|
>
>
>
> **Q2:** This work lacks some necessary baselines, making the experiments unreliable. It is recommended to add the adversarial training method FLAG[3], as well as graph purification
> methods GARNET, ProGNN, and STABLE.
>
> **A2:** We would like to point our that one great advantage of our defense is that its contribution
> is orthogonal to existing defenses. For instance, our BBRW combined with SoftMedian
> exhibits state-of-the-art robustness in our paper. Therefore, adversarial training can be
> applied to our method for further improvement as well.
>
> Moreover, graph purification methods such as ProGNN have been claimed to be much
> weaker than SoftMedian under adaptive attacks in [1]. Therefore, we choose to compare our
> method with this stronger defense (SoftMedian) in our submission.
>
> We also follow your suggestions to include the adversarial training method (GCN-AT) and
> two graph purification methods (GARNET and ProGNN) to further validate our statement.
> The result shows the adversarial trained GCN (GCN-AT) still underperforms our BBRW
> method. In addition, our BBRW method also significantly outperforms GARNET, ProGNN,
> and SoftMedian, which is consistent with the paper [1]. Note that the experiment results in [2] show that
> the robustness of STABLE is worse than RGCN (included in our experiments) across several
> attacks (e.g., Nettack, TDGIA, G-NIA, etc.).
> - [1] Mujkanovic, Felix, et al. ”Are Defenses for Graph Neural Networks Robust?.” Advances
> in Neural Information Processing Systems 35 (2022): 8954-8968.
> - [2] Tao, S., Cao, Q., Shen, H., Wu, Y., Xu, B., & Cheng, X. (2023). IDEA: Invariant Causal
> Defense for Graph Adversarial Robustness. arXiv preprint arXiv:2305.15792.
>
> ### Table: Classification accuracy (%)  (Cora-ML)
>
> | Method         | 0% Clean | 25% Transfer | 25% Adaptive | 50% Transfer | 50% Adaptive | 100% Transfer | 100% Adaptive |
> |----------------|----------|--------------|--------------|--------------|--------------|---------------|---------------|
> | RGCN           | 88.0±6.0 | 72.5±8.4     | 66.0±7.7 | 44.0±8.9     | 36.0±5.4     | 17.5±8.7      | 7.0±4.6       |
> | ProGNN         | 89.0±3.7 | 82.0±9.8     | 76.0±4.9     | 54.0±13.6    | 26.0±10.2    | 26.0±1.2      | 10.0±6.3      |
> | GCN-AT         | 86.0±8.0 | 84.0±10.2    | 74.0±8.9     | 82.0±7.5     | 52.0±7.5     | 80.0±11.0     | 28.0±7.5      |
> | GARNET         | 89.0±3.7 | 66.0±3.8     | 66.0±5.8     | 49.0±6.6     | 38.0±9.8     | 18.0±7.5      | 9.0±4.9       |
> | STABLE        | 89.0±3.7 | 79.0±8.6 | 66.0±8.0 | 47.0±8.7 | 35.0±10.5 | 23.0±6.8|  21.0±3.7|
> | GCN            | 89.5±6.1 | 66.0±9.7     | 66.0±9.7     | 40.5±8.5     | 40.5±8.5     | 12.0±6.4      | 12.0±6.4      |
> | BBRW-GCN       | 90.0±5.5| 89.5±6.1   | 89.0±6.2     | 86.0±5.4 | 85.0±6.3 | 85.0±7.1  | 75.0±10.2 |
> | APPNP          | 90.5±4.7 | 81.5±9.5   | 80.5±10.4   | 66.5±8.7     | 66.0±7.9     | 44.0±9.2      | 43.5±6.4      |
> | BBRW-APPNP     | 91.0±4.9| 89.0±5.4  | 87.5±5.6     | 85.0±7.1     | 83.0±6.4     | 83.5±6.3      | 69.0±9.7      |
> | SoftMedian     | 91.5±5.5 | 86.0±7.0     | 83.0±7.1     | 75.0 ± 8.4 |73.0±7.1     | 48.5±11.4    | 47.5±9.3      |
> | BBRW-SoftMedian| 92.0±4.6| 91.5±5.0  | 92.0±4.6   | 89.5±6.9   | 88.0±5.1   | 87.0±8.4    | 84.5±8.8    |

---

> > ### Author Response · Authors · 2023-11-20
> >
> > **Q3:** The experimental settings are unclear. a) Section 4.1 states, ”We randomly select 20 target
> > nodes per split for robustness evaluation.” If my understanding is correct, does this mean 20
> > nodes are selected per split for training, validation, and testing, totaling 60 nodes? If so,
> > there are 20 nodes from the training set, and is it reasonable to evaluate accuracy on the training set? b) Regarding ”multiple link budgets ∆ ∈ {0%, 25%, 50%, 100%} of the target
> > node’s total degree,” when ∆ = 100%, are the attack still imperceptible? Is it necessary to
> > preserve the original label prediction? c) It would be useful to report the model’s accuracy
> > when the attack perturbation is at 5%, 10%, 15%, and 20%, as done in other studies[5,6].
> >
> > **A3:** We follow the common experiment settings in [1][2], and we are happy to further clarify
> > this setting:
> >
> > a) In the training phase, we use the whole training set to train the model. In the attack phase,
> > we conduct target attacks on the test set and attack one target node every time. We select
> > one target node from the test set every time and repeat this for 20 times, which evaluates 20
> > target nodes in the test set in total. To obtain a reliable result, we conduct the above attack
> > process on 10 random splits of dataset and report the mean and variance over all the splits.
> >
> > b) Actually, our budget setting of 25%, 50% and 100% is common in several works [1][2]
> > and is not contradict to the unnoticeability principle. Specifically, our attack budget setting
> > follows the setting in [1]. Our budget is set according to the target node’s degree. For
> > example, if a node has 5 neighbors, and the budget ∆ = 100%, then the attacker can change
> > up to 5 · 1 = 5 edges. Therefore, we only modify only the unnoticeable number of edges in
> > the whole graph.
> >
> > c) Following your suggestions, we also provide additional experiments under smaller budgets
> > (from 0% to 20%) in the following table. From the result, we observe that under small
> > budgets, our proposed BBRW still outperform other models. The effectiveness become
> > more evident under larger budgets.
> > - [1] Mujkanovic, Felix, et al. ”Are Defenses for Graph Neural Networks Robust?.” Advances
> > in Neural Information Processing Systems 35 (2022): 8954-8968.
> > - [2] Z  ̈ugner, D., Akbarnejad, A., & G  ̈unnemann, S. (2018, July). Adversarial attacks on
> > neural networks for graph data. In Proceedings of the 24th ACM SIGKDD international
> > conference on knowledge discovery & data mining (pp. 2847-2856).
> >
> > #### Table: Classification accuracy (%) under different perturbation rates of graph attack. The best results are in **bold**, and the second-best results are _underlined_. (Cora-ML)
> >
> > | Method   | 0% Clean | 5% Transfer | 5% Adaptive | 10% Transfer | 10% Adaptive | 15% Transfer | 15% Adaptive | 20% Transfer | 20% Adaptive |
> > |----------|----------|-------------|-------------|--------------|--------------|--------------|--------------|--------------|--------------|
> > | MLP      | 73.5±7.4 | 73.5±7.4    | 73.5±7.4    | 73.5±7.4     | 73.5±7.4     | 73.5±7.4     | 73.5±7.4     | 73.5±7.4     | 73.5±7.4     |
> > | MagNet   | 88.5±3.2 | 88.0±3.3    | \           | 86.5±3.9     | \            | 85.0±5.5     | \            | 78.5±8.1     | \            |
> > | RGCN     | 88.0±6.0 | 87.5±6.7    | 87.0±6.8    | 87.0±6.4 | 87.0±6.8   | 85.0±6.7     | 84.5±6.9     | 81.0±8.6     | 77.5±10.3   |
> > | GCN      | 89.5±6.1 | 88.0±6.8    | 88.0±6.8    | 87.5±6.8     | 87.5±6.8     | 83.0±7.1     | 83.0±7.1     | 76.5±9.8     | 76.5±9.8       |
> > | BBRW-GCN | 90.0±5.5 | 90.0±5.5 | 90.0±5.5 | 90.0±5.5 | 89.5±5.2 | 89.5±5.2 | 89.5±5.6 | 89.5±5.7 | 89.0±7.6 |

---

> > > ### Author Response · Authors · 2023-11-20
> > >
> > > **Q4:** The high accuracy of the GCN under 25% adaptive attack does not align with the findings of
> > > the original paper[7]. Please provide performance under the same adaptive attack settings
> > > as in the original paper.
> > >
> > > **A4:** The performance is not consistent with paper [2] due to the difference in the
> > > dataset. Different from the undirected graph datasets used in paper [2], we use the directed
> > > graphs downloaded from the work [1]. We exactly follow the implementation and setting
> > > in [2] and we can guarantee that we evaluate different models fairly under the same attack
> > > setting. We will make our code publicly available after the acceptance to ensure the fair
> > > comparison and reproducibility.
> > > - [1] Zhang, X., He, Y., Brugnone, N., Perlmutter, M., & Hirn, M. (2021). Magnet: A
> > > neural network for directed graphs. Advances in neural information processing systems, 34,
> > > 27003-27015.
> > > - [2] Mujkanovic, Felix, et al. ”Are Defenses for Graph Neural Networks Robust?.” Advances
> > > in Neural Information Processing Systems 35 (2022): 8954-8968.
> > >
> > >
> > > **Q5:** The performance of BBRW is incredible, as mentioned in Section 4.3. What contributes
> > > to its effectiveness?
> > >
> > > **A5:** Thanks for the recognition of our incredible performance. We are happy to explain its
> > > effectiveness as follows.
> > >
> > > First of all, our theoretical analysis in Section 3.3 provides explanations for the effectiveness
> > > of BBRW with proper β value. Essentially, if we can differentiate the roles of in-links
> > > and out-links, we can significantly reduce the impact of attack considering the practical
> > > constraint that attacking the out-links of target nodes is more challenging as explained in the
> > > paper.
> > >
> > > Second, we have included comprehensive ablation studies in Section 4.3 to understand why
> > > BBRW is so incredible and effective. One key advantage of BBRW is that it can defense
> > > against transfer attack and adaptive attack simultaneously. In Figure 5 (a), we can observe
> > > that larger β can perform well under the transfer attack since larger β will reduce the impact
> > > of in-links. But we can not fully trust out-links either since as shown in Figure 5 (b), BBRW
> > > with too large β (such as 1.0) faces a catastrophic performance degradation under adaptive
> > > attack because of the 2-hop indirect attack as shown in Figure 4. Therefore, to balance the
> > > strategy against these two attacks, it is appropriate to specify the β value within the range of
> > > (0.7, 0.8) for BBRW. The simplicity and effectiveness are strong advantages of our method.
> > >
> > > **Q6:** How does BBRW handle node injection attacks[1,2]? In these attacks, only nodes or
> > > edges are injected, with the injected edges being directed. This scenario is practical and
> > > worth exploring.
> > >
> > > **A6:** Our BBRW can also be very effective in this scenario because it can differentiate
> > > the direction of injected edges and significantly mitigate the effect of the directed injected
> > > edges. We use G-NIA in [1] to conduct the injection attack on the backbone models and our
> > > BBRW methods, the following results show the significant effectiveness of our method.
> > >
> > > - [1] Tao, S., Cao, Q., Shen, H., Huang, J., Wu, Y., & Cheng, X. (2021, October). Single
> > > node injection attack against graph neural networks. In Proceedings of the 30th ACM
> > > International Conference on Information & Knowledge Management (pp. 1794-1803).
> > >
> > > ### Table: Graph Injection Attack (Cora-ML)
> > >
> > > | Model        | GCN   | BBRW-GCN | APPNP | BBRW-APPNP |
> > > |--------------|-------|----------|-------|------------|
> > > | Accuracy     | 42.6% | 81.3%    | 61.0% | 84.7%      |
> > >
> > >
> > >
> > > To conclude, we believe we have fully addressed all of your concerns. Please kindly let us know if
> > > you have any further concerns.

---

> > > > ### Author Response · Authors · 2023-11-21
> > > > **A Friendly Reminder**
> > > >
> > > > Dear Reviewer 54vV,
> > > >
> > > > We are thankful for your valuable comments and suggestions. We hope that our responses have addressed all your concerns. As there is only one day remaining in the author-reviewer discussion period, we kindly request you let us know if you have any additional questions. We greatly appreciate your further feedback.
> > > >
> > > > If you find our responses satisfactory, we would kindly request an update of the rating to reflect your assessment. Your feedback is valuable to us, and we appreciate your time and consideration.
> > > >
> > > > Best regards,
> > > >
> > > > All authors

---

> > > > > ### Comment · Reviewer_54vV · 2023-12-05
> > > > > **Reply**
> > > > >
> > > > > Thank the author for their efforts. The newly added experiments A2 and A3 have resolved some of my concerns. However, there is still no intuitive explanation for why the method is effective (Q5), and there is also a lack of explanation for why BBRW performs worse on clean graphs in A1 for both arxiv and pubmed datasets. Taking all factors into consideration, I have changed my score to 5, indicating a weak reject.

---

### Official Review · Reviewer_twAK · 2023-10-31

**Soundness:** 3 good
**Presentation:** 2 fair
**Contribution:** 2 fair
**Rating:** 6
**Confidence:** 4

**Summary:**

The paper studies the adversarial attack and robustness on directed graphs. The authors first propose a simple and more practical setting for attacks on directed graphs which restricted the perturbations on out-links. They conduct experiments with undirected graph neural networks to show there might be a false sense of robustness on directed graphs. To overcome this issue and to enhance the robustness of directed graph, the authors propose a biased bidirectional random walk, which balance the trustworthiness of out-links and in-links with a hyper-parameter $\beta$. They also provide a comprehensive theoretical analysis on the optimal selection of this hyper-parameter. When coupled with the proposed plug-in defense strategy, this framework achieves outstanding clean accuracy and state-of-the-art robust performance against both transfer and adaptive attacks.

**Strengths:**

1. The proposed message-passing framework is a simple and effective approach by leveraging the directional information to enhancing the robustness of GNNs in directed graphs.

2. The paper introduces a new and more realistic directed graph attack setting to overcome the limitations of existing attacks.

3. The paper provides a comprehensive evaluation of the proposed framework and compares it with existing defense strategies on undirected graphs. The experiment results and findings demonstrate that the proposed framework achieves outstanding clean accuracy and state-of-the-art robust performance against both transfer and adaptive attacks.

4. This work presents an innovative approach to GNN attacks, focusing on improving the robustness and trustworthiness of directed graphs.

**Weaknesses:**

1. The description of the attack setting in this paper requires further clarification and precision. If I am understanding correctly, in terms of commonly used terms for adversarial attacks on graphs, this paper focuses on a target attack, while the transfer attack indicates the gray-box attack and adaptive attack is the white box setting. A more explicit definition and distinction between these terms would enhance the reader’s comprehension and align the terminology with established norms in the field.

2. The section discussing catastrophic failures due to indirect attacks seems somewhat disjointed from the existing body of work on directed graphs. The majority of experiments in Table 1 are centered around undirected graph neural networks, leading to a claim of a severe false sense of robustness against transfer attacks in these networks. Given the paper’s earlier assertion that out-links and in-links should be treated distinctly in directed graphs, this leap in logic is perplexing and necessitates a more thorough explanation. A more robust motivation could potentially be achieved by exploring existing attacks on directed graph neural networks, such as DiGNN and MagNet.

3. The proposed attack budget settings, encompassing 25%, 50%, and 100%, appear impractical and neglect the crucial aspect of attack unnoticeability. A more realistic and subtle approach to defining attack budgets would likely yield more applicable and insightful results.

4. The paper seems to lack a discussion on related works specifically addressing attacks or defenses in directed graphs. Clarification is needed as to whether this work stands alone in its focus on directed graph robustness or if there are other relevant studies in this domain.

5. The paper’s approach to conducting adaptive attacks solely on undirected graph neural networks raises questions of fairness and relevance, given the unique characteristics of directed graphs. If the issue stems from challenges related to gradient backpropagation, it would be beneficial to consider relevant baselines, such as Rossi, Emanuele, et al. "Edge Directionality Improves Learning on Heterophilic Graphs." arXiv preprint arXiv:2305.10498 (2023).

Minor Issues:
1. The paper would benefit from the inclusion of explanations for specific notations used, such as $\mathcal{N}$ in section 2.2 on adversarial capacity, and $A_{sym}$ in section 3.1, to aid reader comprehension and provide a more seamless reading experience.

**Questions:**

Please refer to the weakness.

---

> ### Author Response · Authors · 2023-11-20
>
> Dear Reviewer, thanks for your valuable comments and the recognition of the novelty, simplicity,
> and effectiveness of our method. However, we believe there exist some misunderstandings about our
> work, and we are happy to fully address all of your concerns as follows.
>
> **Q1:** The description of the attack setting in this paper requires further clarification and precision.
> If I am understanding correctly, in terms of commonly used terms for adversarial attacks on
> graphs, this paper focuses on a target attack, while the transfer attack indicates the gray-box
> attack and adaptive attack is the white box setting. A more explicit definition and distinction
> between these terms would enhance the reader’s comprehension and align the terminology
> with established norms in the field.
>
> **A1:** We include the details of the attack setting in Section 4.1, and your understanding of the
> attack setting is correct. Let us further clarify them here: (1) This paper mainly focuses on
> target attacks. (2) Transfer attack is gray-box attack, in which the adversarial perturbations
> are transferred from the surrogate model GCN to the specific victim model. Therefore, it
> could be much weaker in many cases. (3) Adaptive attack is white-box attack, and in
> our paper we employ PGD attack to evaluate our model, which is claimed as the strongest
> adaptive attack in [1] and verified by our experiments in Appendix A.2.
> We will improve the clarity in our revision following your suggestions.
>
> - [1] Mujkanovic, Felix, et al. ”Are Defenses for Graph Neural Networks Robust?.” Advances
> in Neural Information Processing Systems 35 (2022): 8954-8968.

---

> > ### Author Response · Authors · 2023-11-20
> >
> > **Q2:** The section discussing catastrophic failures due to indirect attacks seems somewhat dis-
> > jointed from the existing body of work on directed graphs. The majority of experiments
> > in Table 1 are centered around undirected graph neural networks, leading to a claim of a
> > severe false sense of robustness against transfer attacks in these networks. Given the paper’s
> > earlier assertion that out-links and in-links should be treated distinctly in directed graphs,
> > this leap in logic is perplexing and necessitates a more thorough explanation. A more robust
> > motivation could potentially be achieved by exploring existing attacks on directed graph
> > neural networks, such as DiGNN and MagNet.
> >
> > **A2:** Thanks for letting us know your confusion. We believe there are some
> > misunderstandings about the logic that we try to deliver in Section 3.1, and we are happy to
> > clarify it as follows.
> >
> > (1) The reason we did not include the evaluation of existing directed GNNs such as DiGNN
> > and MagNet in Table 1 (Section 3.1) is that we put them into Section 5 (we have to include
> > those there for a complete comparison). Therefore, we remove these evaluations in Section
> > 3.1 to avoid duplicated results. In our revision, we will add a discussion and description
> > in Section 3.1. to refer to these results in Section 4, which will make the logic clear and
> > smoother.
> >
> > (2) In Section 3.1 and Table 1, we mainly focus on providing some insight and motivation
> > for our BBRW algorithm design. Our central claim is that out-links are more reliable than
> > in-links, and therefore we need to differentiable their roles. One naive approach we try
> > is to only use out-links and do not involve in-links, which is exactly what GCN-RW and
> > APPNP-RW achieve in Table 1. But this simple idea will totally fail as we will show in the
> > catastrophic failures under adaptive attack.
> >
> > (3) Following your suggestions, we also include all the directed baselines and our method in
> > the following table to provide a comprehensive overview on the insufficient robustness of
> > existing directed GNNs and the superior effectiveness of our methods. The experimental
> > results demonstrate that: (a) The directed GNNs baselines (DGCN, DiGCN and MagNet)
> > are even not robust under weak transfer attack. (b) The one directional propagated GNNs
> > (GCN-RW and APPNP-RW) are robust under transfer attack but can be easily attacked
> > under stronger adaptive attack (this is what we call catastrophic failures or false sense of
> > robustness). (c) our method outperforms the baseline models significantly across both kinds
> > of attacks and various budgets.
> >
> > ### Table 7: Classification accuracy (%) under transfer and adaptive attacks (Cora-ML)
> >
> > | Method       | 0% Clean | 25% Transfer | 25% Adaptive | 50% Transfer | 50% Adaptive | 100% Transfer | 100% Adaptive |
> > |-------------|----------|--------------|--------------|--------------|--------------|---------------|---------------|
> > | MLP                | 73.5±7.4 | 73.5±7.4     | 73.5±7.4     | 73.5±7.4     | 73.5±7.4     | 73.5±7.4      | 73.5±7.4      |
> > | DGCN                | 64.0±7.0 | 54.0±8.3     | \            | 34.5±10.6    | \            | 27.0±10.1     | \             |
> > | DiGCN               | 66.0±8.6 | 41.5±10.5    | \            | 29.5±8.2     | \            | 21.5±5.9      | \             |
> > | MagNet              | 68.0±6.0 | 51.5±11.2    | \            | 35.0±12.0    | \            | 35.0±7.7      | \             |
> > | GCN                 | 89.5±6.1 | 66.0±9.7     | 66.0±9.7     | 40.5±8.5     | 40.5±8.5     | 12.0±6.4      | 12.0±6.4      |
> > | GCN-RW              | 86.5±6.3 | 86.5±6.3     | 52.0±8.1     | 86.5±6.3     | 28.0±4.6     | 86.5±6.3      | 10.5±5.7      |
> > | APPNP               | 90.5±4.7 | 81.5±9.5     | 80.5±10.4    | 66.5±8.7     | 68.0±12.1    | 44.0±9.2      | 46.0±7.3      |
> > | APPNP-RW            | 85.5±6.5 | 85.5±6.5     | 30.0±7.7     | 85.5±6.3     | 15.0±3.9     | 85.0±6.3      | 11.5±3.2      |
> > | BBRW-GCN (ours)           | 90.0±5.5 | 89.5±6.1     | 89.0±6.2     | 86.0±5.4     | 85.0±6.3     | 85.0±7.1      | 75.0±10.2     |
> > | BBRW-APPNP (ours)  | 91.0±4.9 | 89.0±5.4     | 87.5±5.6     | 85.0±7.1     | 83.0±6.4     | 83.5±6.3      | 69.0±9.7      |

---

> ### Author Response · Authors · 2023-11-20
>
> **Q3:** The proposed attack budget settings, encompassing 25%, 50%, and 100%, appear impractical and neglect the crucial aspect of attack unnoticeability. A more realistic and subtle
> approach to defining attack budgets would likely yield more applicable and insightful results.
>
> **A3:** We would like to point out that evaluating target attack with budgets 25%, 50%,
> and 100% is very common in the literature, such as [1][2]. And the modified edges under
> these budgets are still unnoticeable. Specifically, the budget is set according to the target
> node’s degree. For example, if the target node has 6 neighbors and the budget ∆ = 50%,
> then the attacker can only change up to 6 · 0.5 = 3 edges. Therefore, we only modify a
> unnoticeable number of edges in the whole graph. We are also happy to provide more
> comparison under small budgets following your suggestion (from 0% to 20%). The results
> provide similar phenomenon as the case of 25% budget and also validate the effectiveness
> of our model.
> - [1] Mujkanovic, Felix, et al. ”Are Defenses for Graph Neural Networks Robust?.” Advances
> in Neural Information Processing Systems 35 (2022): 8954-8968.
> - [2] Z  ̈ugner, D., Akbarnejad, A., & G  ̈unnemann, S. (2018, July). Adversarial attacks on
> neural networks for graph data. In Proceedings of the 24th ACM SIGKDD international
> conference on knowledge discovery & data mining (pp. 2847-2856).
>
>
> ### Table: Classification accuracy (%) (Cora-ML)
>
> | Method    | 0% Clean | 5% Transfer | 5% Adaptive | 10% Transfer | 10% Adaptive | 15% Transfer | 15% Adaptive | 20% Transfer | 20% Adaptive |
> |-----------|----------|-------------|-------------|--------------|--------------|--------------|--------------|--------------|--------------|
> | MLP       | 73.5±7.4 | 73.5±7.4    | 73.5±7.4    | 73.5±7.4     | 73.5±7.4     | 73.5±7.4     | 73.5±7.4     | 73.5±7.4     | 73.5±7.4     |
> | MagNet    | 88.5±3.2 | 88.0±3.3    | \           | 86.5±3.9     | \            | 85.0±0.05477 | \            | 78.5±8.1     | \            |
> | RGCN      | 88.0±6.0 | 87.5±6.7    | 87.0±6.8    | 87.0±6.4 | 87.0±6.8   | 85.0±6.7     | 84.5±6.9     | 81.0±8.6     | 77.5±10.3            |
> | GCN       | 89.5±6.1 | 88.0±6.8    | 88.0±6.8    | 87.5±6.8   |  87.5±6.8 | 83.0±7.1     | 83.0±7.1     | 76.5±9.8     | 76.5±9.8               |
> | BBRW-GCN  | 90.0±5.5 | 90.0±5.5    | 90.0±5.5    | 90.0±5.5  | 90.0±5.5     | 89.5±5.2     | 89.5±5.6     | 89.5±5.7     | 89.0±7.6     | \            |
>
>
>
> **Q4:** The paper seems to lack a discussion on related works specifically addressing attacks or
> defenses in directed graphs. Clarification is needed as to whether this work stands alone in
> its focus on directed graph robustness or if there are other relevant studies in this domain.
>
> **A4:** We would like to emphasize that the attack and defense for directed attacks are largely
> unexplored in the literature to the best of our knowledge. Therefore, there is no attack
> specifically designed for directed graphs, not to mention the defenses. This work provides
> the first of such exploration by proposing new settings/algorithms from both attack and
> defense perspectives. Our major discovery is that we can obtain significantly improved
> robustness orthogonal to existing defense techniques by properly exploiting the valuable
> directional information in directed graphs. This points out an new direction and unprece-
> dented opportunities for future research, highlighting the significant contribution of this
> work. We will make further clarifications on this point in our revision.

---

> > ### Author Response · Authors · 2023-11-20
> >
> > **Q5:** The paper’s approach to conducting adaptive attacks solely on undirected graph neural
> > networks raises questions of fairness and relevance, given the unique characteristics of
> > directed graphs. If the issue stems from challenges related to gradient backpropagation, it
> > would be beneficial to consider relevant baselines, such as Rossi, Emanuele, et al. ”Edge
> > Directionality Improves Learning on Heterophilic Graphs.” arXiv preprint arXiv:2305.10498
> > (2023)
> >
> > **A5:** We would like to point out that Transfer Attack provides much weaker attacks than Adaptive
> > Attack, as verified by our experiments and existing works [1]. The existing GNNs designed
> > for directed graphs (such as DGCN, DiGCN and MagNet) are already very weak under
> > Transfer Attacks as shown in Table 2 and Table 3. Therefore, it is not necessary to make
> > them weaker using Adaptive Attacks (white-box). Based on the above arguments, our
> > evaluation, comparison, and conclusion are still valid.
> >
> > - [1] Mujkanovic, Felix, et al. ”Are Defenses for Graph Neural Networks Robust?.” Advances
> > in Neural Information Processing Systems 35 (2022): 8954-8968.
> >
> >
> > ### Table 9: Classification accuracy (%) under transfer and adaptive attacks (Cora-ML)
> >
> > | Method |  | 0% Clean | 25% Transfer | 25% Adaptive | 50% Transfer | 50% Adaptive | 100% Transfer | 100% Adaptive |
> > |--------|--------|----------|--------------|--------------|--------------|--------------|---------------|---------------|
> > | MLP    |        | 73.5±7.4 | 73.5±7.4     | 73.5±7.4     | 73.5±7.4     | 73.5±7.4     | 73.5±7.4      | 73.5±7.4      |
> > | DGCN   |        | 64.0±7.0 | 54.0±8.3     | \            | 34.5±10.6    | \            | 27.0±10.1     | \             |
> > | DiGCN  |        | 66.0±8.6 | 41.5±10.5    | \            | 29.5±8.2     | \            | 21.5±5.9      | \             |
> > | MagNet |        | 68.0±6.0 | 51.5±11.2    | \            | 35.0±12.0    | \            | 35.0±7.7      | \             |
> >
> > To conclude, we believe we have fully addressed all of your concerns. Please kindly let us
> > know if you have any further concerns.

---

> > > ### Author Response · Authors · 2023-11-21
> > >
> > > Dear Reviewer twAK,
> > >
> > > We are thankful for your valuable comments and suggestions. We hope that our responses have addressed all your concerns. As there is only one day remaining in the author-reviewer discussion period, we kindly request you let us know if you have any additional questions. We greatly appreciate your further feedback.
> > >
> > > If you find our responses satisfactory, we would kindly request an update of the rating to reflect your assessment. Your feedback is valuable to us, and we appreciate your time and consideration.
> > >
> > > Best regards,
> > >
> > > All authors

---

### Official Review · Reviewer_kogv · 2023-11-01

**Soundness:** 3 good
**Presentation:** 3 good
**Contribution:** 3 good
**Rating:** 6
**Confidence:** 5

**Summary:**

The authors study adversarial robustness w.r.t. perturbations of the graph structure for directed graphs. They identify a gap in the litreature, namely that most previous robustness studies forcus on undirected graphs. They argue that there is an asymmetry between out-links and in-links and that in some applications it is much easier for an adversary to perturb the in-links compared to the out-links of a target node. They propose RDGA -- a modification of existing attacks with additional restrictions on the out-links of the target node. They also propose a new heuristic defense where we a tunable parameter can place different weights on the in-links and out-links. This change is ortogonal to other defense measures and can be composed with both vanilla GNNs and existing defenses.

**Strengths:**

The simplicity of the proposed defense is in my opinion its biggest strengths. The experiments suggest that there is a range of $\beta$ values for which the clean accuracy is well mantained while showing a boost in adversarial accuracy -- sugestting that we can improve robustness without a significant trade-off.

The experimental analysis is comprehensive. The addition of adaptive attacks is especially appreciated since they are crucial for properly evaluating heuristic defenses.

The theorethical analysis is interesting even though it relies on many simplifying assumptions.

**Weaknesses:**

A big weakness of the paper is that the evaluation is focused on citation networks only -- where for the classification task the out-links are already quite informative. It's not clear how much benefit there could be for other types of networks, where e.g. the patterns of in vs. out-links are different.

The proposed attack can be seen as a relaxation of the even more stringent indirect attack (aslo called the influencer attack by Zugner et al. (2018)) where neither the in-links nor the out-links of the target node can be modified. Therefore, it is not surprusing that the performance is somewhere between the unrestricted and the fully restricted attack. I think elaborating on this connection in the related work is warranted.

Studying the directed setting is imporant, however, whether the attacker is more likely to be able to change the in-links or the out-links depends highly on the application. For example, in social networks it might be true that changing out-links is more difficult but this is not necessarily always the case. Moreover, often there is no "attacker" and we are conducting adversarial robustness studies to quantify e.g. the robustness to worst-case noise -- i.e. treating nature as an adversary. That is to say, I don't think that all future studies should adopt a restricted attack such as the one proposed, but rather include this as another viewpoint.

I think the paper can benefit from further studying the impact of the proposed defense on other aspects of robustness:
- Is the robustness to attribute/feature perturbation positively/negatively affected?
- Does the proposed defense also improved certified robustness (which can be easily tested with one of the black-box randomized smoothing certificates)?
- How is the robustness to global (untargeted) attacks affected?

Essentially the question is what are the trade-offs from adopting the $\beta$ weighted adjacency matrix.

**Questions:**

1. In the ablation study currently the authors break down the links in terms of 1-hop, 2-hop and others. It would be intresting to see the breakdown in terms of in vs. out-links as well, i.e. show: 1-hop (always in-links), 2-hop in-link, 2-hop out-link, etc.
2. How does this approach compare to the more stringent indirect (adaptive) attack?
3. In Table 4 the masking rate starts at 50\%, it would be insightful to also show lower masking rates, and in particular 0\% which would correspond to unrestricted attacks. This will help in understanding whether the proposed defenses is "universally" helpful for different threat models.
4. It should be straightforward to set different $\beta$ values for each node, rather than a single global value. Tuning can be avoided by setting these values based on the theory.

---

> ### Author Response · Authors · 2023-11-20
>
> Dear reviewer, thank you so much for your valuable comments and your recognition of the simplicity and effectiveness of our method, our comprehensive experiments, and the theoretical analysis. We are happy to fully address all of your concerns as follows.
>
> **Q1:**  A big weakness of the paper is that the evaluation is focused on citation networks only –
> where for the classification task the out-links are already quite informative. It’s not clear
> how much benefit there could be for other types of networks, where e.g. the patterns of in vs.
> out-links are different.
>
> **A1:** Thank you for your suggestion! Generally speaking, both the out-links and in-links can be informative when considering the robustness of GNNs. Our proposed BBRW leverages both the out-links and in-links information and has the flexibility to adjust our trust in them through hyperparameter β. Therefore, our method is generally effective for other networks.
>
> Besides citation networks, we also include the wikipedia-based relational network called WikiCS. We use the dataset in [1] and follow the dataset preprocessing setting in [2]. We perform the experiments on WikiCS across various budgets and report the result in the following table. The following results show the significant robustness of BBRW and validate that our BBRW can be generalized to different types of networks beyond citation networks.
>
> References:
> - [1] Péter Mernyei and Catalina Cangea. Wiki-cs: A wikipedia-based benchmark for graph neural networks. arXiv preprint arXiv:2007.02901, 2020.
> - [2] Zhang, X., He, Y., Brugnone, N., Perlmutter, M., & Hirn, M. (2021). Magnet: A neural network for directed graphs. Advances in neural information processing systems, 34, 27003-27015.
>
> ### Table : Classification accuracy (%) under different perturbation rates of graph attack. (WikiCS)
>
> | Method   | Clean (0%)    | Transfer (25%)  | Adaptive (25%) | Transfer (50%) | Adaptive (50%) | Transfer (100%)  | Adaptive (100%) |
> |----------|-----------|-----------|----------|-----------|----------|-----------|----------|
> | MLP      | 44.0±9.2  | 44.0±9.2  | 44.0±9.2 | 44.0±9.2  | 44.0±9.2 | 44.0±9.2  | 44.0±9.2 |
> | MagNet   | 64.0±5.8  | 40.0±3.2  | \        | 29.0±7.3  | \        | 25.0±7.7  | \        |
> | RGCN     | 70.0±4.5  | 45.0±11.8 | 39.0±9.7 | 36.0±12.4 | 23.0±12.9| 22.0±6.0  | 17.0±9.8 |
> | GCN      | 70.0±8.4  | 39.0±12.4 | 39.0±12.4| 23.0±8.7  | 23.0±8.7 | 16.0±8.6  | 16.0±8.6 |
> | BBRW-GCN | 70.0±8.3  | 62.0±9.3  | 61.0±10.7| 55.0±10.5 | 49.0±12.4| 51.0±13.2 | 39.0±13.2|
>
> **Q2:** The proposed attack can be seen as a relaxation of the even more stringent indirect attack
> (also called the influencer attack by Zugner et al. (2018)) where neither the in-links nor
> the out-links of the target node can be modified. Therefore, it is not surprising that the
> performance is somewhere between the unrestricted and the fully restricted attack. I think
> elaborating on this connection in the related work is warranted. How does this approach
> compare to the more stringent indirect (adaptive) attack?
>
> **A2:** Thank you for your thoughtful comments. Exactly, our proposed attack RDGA can
> be seen as a bridge between direct attack and indirect attack. As shown in Figure 3, the attacker
> tries to make a decision to maximize the attack strength between in-link direct attack and
> indirect attack. Therefore, RDGA is stronger than the indirect (adaptive) attack. To verify
> this conclusion, we also provide the comparison of the strengths of RDGA and Indirect
> adaptive attack as follows. The comparison indicates that the proposed RDGA attack is
> indeed stronger than the Indirect attack setting by Zugner et al (2018) as you suggest. We
> will discuss this connection in our revision.
>
> ### Table: Classification accuracy (%) of BBRW-GCN (Citeseer)
>
> |  Budget        | 0%        | 25%       | 50%       | 100%      |
> |----------|-----------|-----------|-----------|-----------|
> | RDGA     | 61.5±7.4  | 43.0±10.3 | 27.0±14.4 | 20.5±9.6  |
> | Indirect | 61.5±7.4  | 60.0±8.4  | 46.0±10.7 | 35.0±7.1  |

---

> > ### Author Response · Authors · 2023-11-20
> >
> > **Q3:** Studying the directed setting is important, however, whether the attacker is more likely to
> > be able to change the in-links or the out-links depends highly on the application. For
> > example, in social networks, it might be true that changing out-links is more difficult but
> > this is not necessarily always the case. Moreover, often there is no ”attacker” and we are
> > conducting adversarial robustness studies to quantify e.g. the robustness to worst-case noise
> > – i.e. treating nature as an adversary. That is to say, I don’t think that all future studies
> > should adopt a restricted attack such as the one proposed, but rather include this as another
> > viewpoint.
> >
> > **A3:** This is a very thoughtful comment. Indeed, we cannot argue that it is more likely to be able to change the in-links or the out-links for all applications. However, our BBRW method provides such flexibility to be easily customized for specific applications: (1) if it is easier to attack in-links, we can choose β > 0.5; (2) if it is easier to attack out-links, we can choose β < 0.5; (3) we can even choose β for each node depending on the prior knowledge of the trustworthiness of local links. We will make further clarifications on this point in our revision.
> >
> > We would like to emphasize that the attack and defense for directed attacks are largely unexplored in the literature to the best of our knowledge. Therefore, there is no attack specifically designed for directed graphs, not to mention the defenses. This work provides the first of such exploration by proposing new settings/algorithms from both attack and defense perspectives. Our major discovery is that we can obtain significantly improved robustness orthogonal to existing defense techniques by properly exploiting the valuable directional information in directed graphs. This points out a new direction and unprecedented opportunities for future research, highlighting the significant contribution of this work.
> >
> >
> > **Q4:** Is the robustness to attribute/feature perturbation positively/negatively affected?
> >
> > **A4:** The attribute/feature perturbation would surely affect the robustness since the
> > focus of our work is on the robustness against the structure perturbation. However, compared
> > with the baselines, the robustness to attribute/feature perturbation will not be negatively
> > affected in our new approach.
> >
> > To verify this, we provide the comparison of feature perturbation across different budgets on
> > Cora-ML dataset. The following experimental results show that the feature perturbation can
> > decrease the performance of both GCN and BBRW-GCN. But our BBRW method is still
> > effective and significantly outperforms the backbone model under this attack.
> >
> > ### Table: Classification accuracy (%) under different perturbation rates of graph attack.  (Cora-ML)
> >
> > |                                | 0%       | 50%      | 100%     |
> > |--------------------------------|----------|----------|----------|
> > | GCN                            | 89.5±6.1 | 40.5±8.5 | 12.0±6.4 |
> > | BBRW-GCN                   | 90.0±5.5 | 86.0±5.4 | 85.0±7.1 |
> > | GCN w/ attribute perturbation  | 87.0±5.1 | 26.0±8.0 | 2.0±2.4  |
> > | BBRW-GCN w/ attribute perturbation | 89.0±4.9 | 72.3±4.4 | 70.0±5.5 |

---

> > > ### Author Response · Authors · 2023-11-20
> > >
> > > **Q5:** Does the proposed defense also improve certified robustness (which can be easily tested
> > > with one of the black-box randomized smoothing certificates)?
> > >
> > > **A5:** Thank you for this great suggestion. Although we believe we already use the strongest attack
> > > to evaluate the robustness of our submission, we also agree that certified robustness can
> > > provide a more reliable and theoretically guaranteed evaluation regardless of the attacks.
> > > Following your suggestion, we leverage the randomized smoothing on GNNs in Wang et al
> > > (2021) [1] to evaluate the certified accuracy. The following experimental results show that
> > > our BBRW-GCN can significantly outperform the baseline in terms of certified robustness,
> > > which further verifies our improvements. We will incorporate a more comprehensive
> > > evaluation in our revision.
> > >
> > > ### Table: Certified accuracy (%) (Cora-ML)
> > >
> > > | Perturbation Size | 0      | 1      | 2      | 3  | 4  |
> > > |-------------------|--------|--------|--------|----|----|
> > > | GCN               | 61.39% | 59.18% | 38.78% | 0% | 0% |
> > > | BBRW-GCN          | 83.67% | 81.63% | 73.47% | 59.39% | 0% |
> > >
> > > Reference:
> > > - [1] Wang, B., Jia, J., Cao, X., & Gong, N. Z. (2021, August). Certified robustness of graph neural networks against adversarial structural perturbation. In Proceedings of the 27th ACM SIGKDD Conference on Knowledge Discovery & Data Mining (pp. 1645-1653).
> > >
> > >
> > > **Q6:** How is the robustness to global (untargeted) attacks affected?
> > >
> > > **A6:** Yes, our method can also improve the robustness under global attacks. We provide
> > > additional experiments under the global attack. The following experimental results also
> > > validate that our BBRW method still significantly outperforms the GCN backbone in terms
> > > of robustness under global attacks. The improvement is more significant under the larger
> > > budget.
> > >
> > > ### Table: Classification accuracy (%) under different perturbation rates of graph attack. (Global)
> > >
> > > | Budget | 0%     | 5%     | 10%    | 15%    | 25%    | 50%    | 100%   |
> > > |--------|--------|--------|--------|--------|--------|--------|--------|
> > > | GCN    | 81.7±1.7 | 72.3±1.8 | 66.2±1.4 | 61.9±1.5 | 54.6±1.9 | 39.9±3.0 | 20.6±3.2 |
> > > | BBRW-GCN | 81.5±1.4 | 74.1±1.3 | 70.7±1.8 | 68.3±1.7 | 64.1±2.1 | 56.3±2.3 | 47.7±1.8 |
> > >
> > >
> > > **Q7:** In the ablation study currently the authors break down the links in terms of 1-hop, 2-hop
> > > and others. It would be interesting to see the breakdown in terms of in vs. out-links as well,
> > > i.e. show: 1-hop (always in-links), 2-hop in-link, 2-hop out-link, etc.
> > >
> > >
> > > **A7:** Thanks for the nice suggestion. Actually, the 2-hop-in-link attack is very weak and only
> > > occupies a tiny part of the perturbations. Here is the more detailed breakdown of adversarial
> > > perturbations:
> > > - Transfer: 1-hop-in (96.32%), 2-hop-out (1.23%), 2-hop-in (0.28%), others (2.17%).
> > > - Adaptive: 1-hop-in (71.60%), 2-hop-out (14.01%), 2-hop-in (0.85%), others (13.54%).

---

> > > > ### Author Response · Authors · 2023-11-20
> > > >
> > > > **Q8:** In Table 4 the masking rate starts at 50%, it would be insightful to also show lower masking
> > > > rates, and in particular 0% which would correspond to unrestricted attacks. This will help
> > > > in understanding whether the proposed defenses is ”universally” helpful for different threat
> > > > models.
> > > >
> > > > **A8:** Thanks for the suggestions.
> > > > Following your suggestion, we also provide additional experiments on the effect of lower
> > > > masking rates. The comparison in the following table shows that (1) when the masking rate
> > > > is 0% (unrestricted attacks), our BBRW is almost the same as the backbone GCN model;
> > > > (2) when the mask rate increases (RDGA setting), BBRW outperforms the backbone GCN
> > > > model; (3) when the mask rate is larger, the advantage of our BBRW method is stronger.
> > > > These verify that the proposed defense is ”universally helpful for different threat models.
> > > >
> > > > ### Table: Ablation study on masking rates of target nodes’ out-links under adaptive attack (Cora-ML).
> > > >
> > > > | Masking Rate | 0%      | 10%     | 20%     | 30%     | 40%     |
> > > > |--------------|---------|---------|---------|---------|---------|
> > > > | GCN          | 40.5±8.5 | 40.5±8.5 | 40.5±8.5 | 40.5±8.5 | 40.5±8.5 |
> > > > | BBRW-GCN     | 41.0±8.6 | 45.0±9.7 | 49.0±9.2 | 49.0±10.7| 50.0±11.2|
> > > >
> > > > | Masking Rate | 50%     | 60%     | 70%     | 80%     | 90%     | 100%    |
> > > > |--------------|---------|---------|---------|---------|---------|---------|
> > > > | GCN          | 40.5±8.5 | 40.5±8.5 | 40.5±8.5 | 40.5±8.5 | 40.5±8.5 | 40.5±8.5 |
> > > > | BBRW-GCN     | 52.0±11.4| 54.5±10.8| 56.5±9.2 | 60.0±10.4| 60.5±11.0| 85.0±6.3 |
> > > >
> > > > **Q9:** It should be straightforward to set different β values for each node, rather than a single
> > > > global value. Tuning can be avoided by setting these values based on the theory.
> > > >
> > > > **A9:** This is a very thoughtful viewpoint! Our experimental result shows that only
> > > > setting a global value can already contribute to a significant improvement. This simplicity
> > > > and effectiveness are also big advantages of our method. Setting different β for every single
> > > > node could be more effective, and we would like to explore this brilliant idea in future work.
> > > >
> > > > To conclude, we believe we have fully addressed all of your concerns. Thanks again for
> > > > your support and thoughtful suggestions. Please kindly let us know if you have any further concerns.

---

> > > > > ### Author Response · Authors · 2023-11-21
> > > > >
> > > > > Dear Reviewer kogv,
> > > > >
> > > > > We are thankful for your valuable comments and suggestions. We hope that our responses have addressed all your concerns. As there is only one day remaining in the author-reviewer discussion period, we kindly request you let us know if you have any additional questions. We greatly appreciate your further feedback.
> > > > >
> > > > > If you find our responses satisfactory, we would kindly request an update of the rating to reflect your assessment. Your feedback is valuable to us, and we appreciate your time and consideration.
> > > > >
> > > > > Best regards,
> > > > >
> > > > > All authors

---

> > > > > > ### Comment · Reviewer_kogv · 2023-11-23
> > > > > > **Reply**
> > > > > >
> > > > > > Thank you for the detailed response. I keep my score in favor of acceptance.

---

### Meta-Review · Area_Chair_HZQq · 2023-12-05

**Metareview:**

The authors study adversarial robustness w.r.t. perturbations of the graph structure for directed graphs. They identify a gap in the litreature, namely that most previous robustness studies forcus on undirected graphs. They argue that there is an asymmetry between out-links and in-links and that in some applications it is much easier for an adversary to perturb the in-links compared to the out-links of a target node. They propose RDGA -- a modification of existing attacks with additional restrictions on the out-links of the target node. They also propose a new heuristic defense where we a tunable parameter can place different weights on the in-links and out-links. This change is ortogonal to other defense measures and can be composed with both vanilla GNNs and existing defenses. Specifically, the strength of this paper includes several aspects. 1) The simplicity of the proposed defense is in my opinion its biggest strengths. The experiments suggest that there is a range of values for which the clean accuracy is well mantained while showing a boost in adversarial accuracy -- sugestting that we can improve robustness without a significant trade-off. 2) The experimental analysis is comprehensive. The addition of adaptive attacks is especially appreciated since they are crucial for properly evaluating heuristic defenses. 3) The theorethical analysis is interesting even though it relies on many simplifying assumptions.

However, there are several points to be further improved. For example, this paper lacks empirical evaluations on larger datasets, such as ogb datasets or reddit, which makes us concerned about usefulness on large networks. This work lacks some necessary baselines, making the experiments unreliable. The experimental settings are unclear. The high accuracy of the GCN under 25% adaptive attack does not align with the findings of the original paper. Please provide performance under the same adaptive attack settings as in the original paper. Therefore, this paper cannot be accepted at ICLR this time, but the enhanced version is highly encouraged to submit other top-tier venues.

**Justification For Why Not Higher Score:**

However, there are several points to be further improved. For example, this paper lacks empirical evaluations on larger datasets, such as ogb datasets or reddit, which makes us concerned about usefulness on large networks. This work lacks some necessary baselines, making the experiments unreliable. The experimental settings are unclear. The high accuracy of the GCN under 25% adaptive attack does not align with the findings of the original paper. Please provide performance under the same adaptive attack settings as in the original paper. Therefore, this paper cannot be accepted at ICLR this time, but the enhanced version is highly encouraged to submit other top-tier venues.

**Justification For Why Not Lower Score:**

However, there are several points to be further improved. For example, this paper lacks empirical evaluations on larger datasets, such as ogb datasets or reddit, which makes us concerned about usefulness on large networks. This work lacks some necessary baselines, making the experiments unreliable. The experimental settings are unclear. The high accuracy of the GCN under 25% adaptive attack does not align with the findings of the original paper. Please provide performance under the same adaptive attack settings as in the original paper. Therefore, this paper cannot be accepted at ICLR this time, but the enhanced version is highly encouraged to submit other top-tier venues.

---

### Decision · Program_Chairs · 2024-01-16

Reject